# DAGs with No Fears: A Closer Look at Continuous Optimization for Learning Bayesian Networks

**Dennis Wei**
IBM Research
dwei@us.ibm.com

**Tian Gao**
IBM Research
tgao@us.ibm.com

**Yue Yu**
Lehigh University
yuy214@lehigh.edu

## Abstract

This paper re-examines a continuous optimization framework dubbed NOTEARS for learning Bayesian networks. We first generalize existing algebraic characterizations of acyclicity to a class of matrix polynomials. Next, focusing on a one-parameter-per-edge setting, it is shown that the Karush-Kuhn-Tucker (KKT) optimality conditions for the NOTEARS formulation cannot be satisfied except in a trivial case, which explains a behavior of the associated algorithm. We then derive the KKT conditions for an equivalent reformulation, show that they are indeed necessary, and relate them to explicit constraints that certain edges be absent from the graph. If the score function is convex, these KKT conditions are also sufficient for local minimality despite the non-convexity of the constraint. Informed by the KKT conditions, a local search post-processing algorithm is proposed and shown to substantially and universally improve the structural Hamming distance of all tested algorithms, typically by a factor of 2 or more. Some combinations with local search are both more accurate and more efficient than the original NOTEARS.

## 1 Introduction

Bayesian networks are directed probabilistic graphical models used to model joint probability distributions of data in many applications [20, 25]. Automatic discovery of their directed acyclic graph (DAG) structure is important to research areas from causal inference to biology. However, DAG structure learning is in general an NP-hard problem [8]. Many learning algorithms have been proposed to circumvent exhaustive search in the discrete space of DAGs, including those for discrete variables [7, 1, 24, 16, 9, 30, 12] and continuous variables [6, 27].

Recently, Zheng et al. [31] proposed a *continuous* optimization formulation, referred to as NOTEARS, in which acyclicity of the graph is enforced by a trace of matrix exponential constraint on a weighted adjacency matrix. Several works have since successfully extended the formulation to nonlinear and nonparametric models [29, 19, 17, 32].

This paper takes further steps toward fulfilling the promise of [31] in opening the door to continuous optimization techniques for score-based structure learning. We contribute in particular to theoretical understanding of this framework, leading to significant algorithmic improvements.

First, in Section 2, the acyclicity constraints of [31, 29] are generalized to a class of matrix polynomials with positive coefficients whose traces characterize acyclicity. We also provide a characterization involving the gradient of functions in this class, which is not only essential to proving later results but also has an intuitive graphical interpretation.

In Section 3.1, we revisit the NOTEARS formulation of [31] in which a weighted adjacency matrix is obtained by element-wise squaring of the parameter matrix. It is shown that the Karush-Kuhn-Tucker (KKT) optimality conditions for this constrained optimization cannot be satisfied except in a trivial case. This negative result is somewhat surprising given the empirical success of the augmented

Lagrangian algorithm of [31], and we use the result to explain why the algorithm does not converge to an exactly acyclic solution even when the penalty parameters are very high.

In Section 3.2, we consider an equivalent reformulation in which the adjacency matrix is given by the absolute value of the parameter matrix, motivated in part by the connection between the $\ell_1$ norm and sparsity. We show that the KKT conditions for this reformulation are indeed necessary conditions of optimality, i.e. they are satisfied by all local minima, although even here common constraint qualification methods turn out to fail. If the score function is convex, then the KKT conditions are also sufficient for local minimality, despite the non-convexity of the constraint. We then relate the KKT conditions to the optimality conditions for score optimization subject to explicit edge absence constraints. The KKT conditions can thus be understood through edge absences: together these must be sufficient to ensure acyclicity, but each absence must also be necessary in preventing a cycle.

The theoretical development of Section 3.2 naturally suggests two algorithms: an augmented La-grangian algorithm as in [31] with an absolute value adjacency matrix instead of quadratic, and a local search algorithm, KKTS, informed by the KKT conditions and proven to satisfy them. We find in Section 5 that neither of these algorithms yields state-of-the-art accuracy by itself. However, when combined with other algorithms, KKTS substantially reduces structural Hamming distance (SHD) with respect to the true graph, typically by a factor of at least 2. Moreover, this improvement is consistent across dimensions and base algorithms. In the case of NOTEARS, new state-of-the-art accuracy is obtained, while other combinations can outperform NOTEARS and take less time.

**More on related work**   Bayesian network structure learning has long been an active research area. Constraint- and score-based methods utilize independence tests and graph scores respectively to learn the DAG structure. Optimization methods such as greedy search [7], dynamic programming [18], branch and bound [10], A* search [30, 28], local-to-global search [13] as well as approximation methods [22] have all been proposed. As mentioned, this paper is most closely related to the continuous framework of [31] and subsequent works [29, 32]. Regression-based methods for DAG learning, without the matrix exponential constraint, have also been carefully studied [23, 6, 2, 14].

## 2   Characterizations of acyclicity

In this first section, we provide algebraic characterizations of acyclicity for a directed graph in terms of its adjacency matrix. For a directed graph $\mathcal{G} = (\mathcal{V}, \mathcal{E})$ with vertices $\mathcal{V} = \{1, \ldots, d\}$ and directed edges $(i, j) \in \mathcal{E}$, a non-negative matrix $A$ is a (weighted) adjacency matrix for $\mathcal{G}$ if $A_{ij} > 0$ for $(i, j) \in \mathcal{E}$ and $A_{ij} = 0$ otherwise.

We consider a class of functions $h(A)$ corresponding to matrix polynomials of degree $d$ with positive coefficients, $P(A) = c_0 I + c_1 A + \cdots + c_d A^d$ with $c_p > 0$ for $p = 1, \ldots, d$, from which we define

$$h(A) = \operatorname{tr}(P(A)) - c_0 d = \sum_{p=1}^{d} c_p \operatorname{tr}(A^p). \tag{1}$$

This class includes the function $h(A) = \operatorname{tr}\big((I + A/d)^d\big) - d$ from [29], which corresponds to $c_p = \binom{d}{p}/d^p$, and the trace of matrix exponential from [31],

$$h(A) = \operatorname{tr}(e^A) - d = \sum_{p=1}^{\infty} \frac{\operatorname{tr}(A^p)}{p!}. \tag{2}$$

Although (2) appears to be an infinite power series, it can be rewritten as a finite series with no powers higher than $d$ using the Cayley-Hamilton theorem [15], which equates $A^d$ to a linear combination of $I, A, \ldots, A^{d-1}$, and similarly for all higher powers of $A$.

Any function $h(A)$ in (1) can characterize acyclicity. We defer all proofs to the supplement (SM).

**Theorem 1.** *A directed graph $\mathcal{G}$ is acyclic if and only if its (weighted) adjacency matrix satisfies $h(A) = 0$ for any $h$ defined by* (1).

The proof of Theorem 1 is facilitated by Lemma 1 below. We recall that a matrix $B$ is said to be *nilpotent* if $B^p = 0$ for some power $p \in \mathbb{N}$, or equivalently if $\operatorname{tr}(B^p) = 0$ for all $p \in \mathbb{N}$ [15]. We state the lemma here as there may be independent interest in alternative ways of enforcing nilpotency.

**Lemma 1.** *A directed graph $\mathcal{G}$ is acyclic if and only if its (weighted) adjacency matrix $A$ is nilpotent.*

The gradient of $h(A)$ in (1) is a matrix-valued function given by $\nabla h(A) = \sum_{p=1}^{d} p c_p \left( A^{p-1} \right)^T$. Off-diagonal elements $(\nabla h(A))_{ij}$ have an intuitive interpretation in terms of *directed walks* from $j$ to $i$, i.e. a sequence of edges $(j, i_1), (i_1, i_2), \dots, (i_{l-1}, i) \in \mathcal{E}$. If there is a directed walk from $j$ to $i$, then there is also a *directed path*, i.e. a directed walk in which all vertices $j, i_1, \dots, i_{l-1}, i$ are distinct [5].

**Lemma 2.** *For any $h(A)$ defined by* (1) *and $i \neq j$, $(\nabla h(A))_{ij} > 0$ if and only if there exists a directed walk from $j$ to $i$ in $\mathcal{G}$.*

The gradient $\nabla h(A)$ can also be used to characterize acyclicity, which will prove useful in the sequel.

**Lemma 3.** *A directed graph $\mathcal{G}$ is acyclic if and only if the Hadamard product $A \circ \nabla h(A) = 0$ for any $h$ defined by* (1).

With the help of Lemma 2, we can give a simple graphical interpretation of Lemma 3: If a directed graph is acyclic, then for every pair $(i, j)$, we must either not have an edge from $i$ to $j$, i.e. $A_{ij} = 0$, or not have a return path from $j$ to $i$, i.e. $(\nabla h(A))_{ij} = 0$.

## 3 Analysis of continuous acyclicity-constrained optimization

In the remainder of the paper, we address the problem of learning a Bayesian network (a probabilistic directed graphical model) for the joint distribution of a $d$-dimensional random vector $X$, given a data matrix of $n$ samples $\mathbf{X} \in \mathbb{R}^{n \times d}$. We assume that the Bayesian network is parametrized by a matrix $W \in \mathbb{R}^{d \times d}$ such that the sparsity pattern of $W$ corresponds to the adjacency pattern of the graph: $W_{ij} \neq 0$ if and only if $(i, j) \in \mathcal{E}$. In other words, each edge is associated with a single parameter $W_{ij}$. The most straightforward instance of this setting is a linear structural equation model (SEM) given by $X_j = W_{\cdot j}^T X + z_j$, where $W_{\cdot j}$ is the $j$th column of $W$ and $z_j$ is random noise. More general models such as generalized linear models $\mathbb{E}\left[X_j \mid X\right] = g\left(W_{\cdot j}^T X\right)$ are also included. While we experiment only with continuous variables in Section 5, it is straightforward to accommodate binary variables as well: in a generalized linear structural equation, a single parameter $W_{ij}$ can account for the effect of a binary input variable $X_i$, while a suitable link function $g$ (e.g. logistic) can be used for a binary output $X_j$.

This section analyzes the continuous optimization problem of minimizing a score function $F(W)$ subject to the acyclicity constraint $h(A) = 0$ for any $h$ defined by (1) (thanks to Theorem 1). For simplicity, it is assumed in this section that $F(W)$ is continuously differentiable, although it is not hard to extend the analysis to account for an $\ell_1$ penalty as in (11). We consider two ways of defining a weighted adjacency matrix $A$ from $W$. Section 3.1 re-examines the quadratic case $A = W \circ W$ proposed in [31], while Section 3.2 studies the absolute value case $A = |W|$.

### 3.1 Quadratic adjacency matrix

With $A = W \circ W$ as the element-wise square of $W$, the optimization problem is

$$\min_{W} \quad F(W) \quad \text{s.t.} \quad h(W \circ W) \leq 0. \tag{3}$$

The constraint $h(W \circ W) \leq 0$ is equivalent to $h(W \circ W) = 0$ because $h(A) \geq 0$ for non-negative $A$, as seen from (1). The matrix exponential case of (3) with $h(A)$ as in (2) was proposed in [31].

Applying Lemma 3 yields the following consequence.

**Lemma 4.** *Let $W$ be a feasible solution to problem* (3). *Then $\nabla_W(h(W \circ W)) = 0$.*

The vanishing gradient in Lemma 4 has theoretical and practical implications. First, the Karush-Kuhn-Tucker (KKT) conditions of optimality [4] for problem (3), namely

$$\nabla F(W) + \lambda \nabla_W(h(W \circ W)) = 0 \tag{4}$$

with Lagrange multiplier $\lambda \geq 0$, are not satisfied for any feasible solution except in a trivial case.

**Proposition 2.** *Let $W$ be a feasible solution to problem* (3). *Then unless $W$ is an unconstrained stationary point of $F(W)$, i.e. $\nabla F(W) = 0$, the KKT condition* (4) *cannot hold.*

In particular if $F(W)$ is convex, the condition $\nabla F(W) = 0$ holds only for unconstrained minimizers of $F(W)$, so if these solutions are already acyclic, there is nothing more to be done.

On the practical side, Lemma 4 sheds light on the augmented Lagrangian algorithm proposed in [31]. The augmented Lagrangian corresponding to (3) with penalty parameters $\alpha$ and $\rho$ is

$$F(W) + \alpha h(W \circ W) + \frac{\rho}{2} h(W \circ W)^2, \tag{5}$$

with gradient $\nabla F(W) + (\alpha + \rho h(W \circ W))\nabla_W(h(W \circ W))$.

**Proposition 3.** *Let $W$ be a feasible solution to problem* (3). *Then unless $W$ is an unconstrained stationary point of $F(W)$, $W$ cannot be a stationary point of the augmented Lagrangian* (5).

Proposition 3 explains the following observed behavior of the augmented Lagrangian algorithm, namely that it does not converge to an exactly (or within machine precision) feasible solution of (3) even when the penalty parameters $\alpha$, $\rho$ are very high ($\rho \sim 10^{16}$). The reason is that a minimizer of the augmented Lagrangian (5) cannot be a feasible solution to (3) except in the trivial case discussed above. However, when $\alpha$ and $\rho$ are very large, minimizers of (5) do tend to have gradients $\nabla_W(h(W \circ W)) \approx 0$, and accordingly $h(W \circ W) \approx 0$ by continuity. Thus as $\alpha$ and $\rho$ increase, the augmented Lagrangian algorithm yields solutions that are closer and closer to being feasible.

### 3.2 Absolute value adjacency matrix

As an alternative, we turn to the absolute value definition $A = |W|$. The problem becomes

$$\min_{W} \quad F(W) \quad \text{s.t.} \quad h(|W|) \le 0. \tag{6}$$

Formulation (6) is motivated in part by the failure to satisfy KKT conditions in Section 3.1 and in part by the connection between the absolute value function/$\ell_1$ norm and sparsity, which is needed for acyclicity. While it will be seen that (6) has different theoretical and numerical properties from (3), the two formulations are equivalent in a sense because acyclicity depends only on the sparsity pattern of $W$, which is clearly the same regardless of whether $|W|$ or $W \circ W$ is used.

**An equivalent smooth optimization** Problem (6) is not a smooth optimization because of the absolute value function. To avoid any issues with continuous differentiability, we make use of the following alternative formulation, which we show in the SM to be equivalent to (6):

$$\min_{W^+, W^-} \quad F\left(W^+ - W^-\right) \quad \text{s.t.} \quad h\left(W^+ + W^-\right) \le 0, \quad W^+, W^- \ge 0. \tag{7}$$

Given any solution $(W^+, W^-)$ to (7), a solution to (6) is obtained simply as $W = W^+ - W^-$.

#### 3.2.1 KKT conditions and constraint qualification

We proceed to analyze the KKT conditions for the smooth reformulation (7), which are as follows:

$$\pm\nabla F\left(W^+ - W^-\right) + \lambda\nabla h\left(W^+ + W^-\right) = M^\pm \ge 0 \tag{8a}$$

$$W^\pm \circ M^\pm = 0, \tag{8b}$$

in addition to the feasibility conditions in (7). The $\pm$ versions of (8a) result from taking gradients with respect to $W^+$ and $W^-$ respectively, where $\lambda \ge 0$ is a Lagrange multiplier. $M^+$, $M^-$ are non-negative matrices of Lagrange multipliers corresponding to the non-negativity constraints in (7), with complementary slackness conditions (8b).

As in Section 3.1, we must consider whether the KKT conditions are *necessary* conditions of optimality, i.e. whether a local minimum must satisfy them. Theorem 6 gives an affirmative answer; however, it turns out that common *constraint qualifications* used to establish necessity do not hold. We refer to [4] and the SM for definitions of regularity and quaisnormality below.

**Proposition 4.** *A feasible solution $(W^+, W^-)$ to problem* (7) *cannot be regular.*

**Proposition 5.** *A feasible solution $(W^+, W^-)$ to problem* (7) *cannot be quasinormal.*

In spite of these negative results, the SM provides a direct proof of the necessity of the KKT conditions (8). The proof uses the following lemma, which we highlight because of its graphical interpretation in terms of directed paths not being created/destroyed by the addition/removal of certain edges.

**Lemma 5.** *For a non-negative matrix $A$, if $(\nabla h(A))_{ij} > 0$, changing the values of $A_{kj}$ for any $k$ cannot make $(\nabla h(A))_{ij} = 0$. Similarly if $(\nabla h(A))_{ij} = 0$, changing the values of $A_{kj}$ for any $k$ cannot make $(\nabla h(A))_{ij} > 0$.*

**Theorem 6.** *Let $(W^+, W^-)$ be a local minimum of problem* (7). *Then there exist a Lagrange multiplier $\lambda \geq 0$ and matrices $M^+ \geq 0$, $M^- \geq 0$ satisfying the KKT conditions in* (8).

### 3.2.2 Relationships with explicit edge absence constraints

We now discuss relationships between the KKT conditions (8) and the optimality conditions for score optimization problems with explicit edge absence constraints, which correspond to zero-value constraints on the matrix $W$. Given a set $\mathcal{Z}$ of such constraints, we consider the problem

$$\min_{W} \ F(W) \quad \text{s.t.} \quad W_{ij} = 0, \quad (i,j) \in \mathcal{Z} \tag{9}$$

and denote by $W^*(\mathcal{Z})$ an optimal solution. The necessary conditions of optimality for (9) are

$$(\nabla F(W))_{ij} = 0, \quad (i,j) \notin \mathcal{Z}, \qquad W_{ij} = 0, \quad (i,j) \in \mathcal{Z}. \tag{10}$$

In one direction, given a KKT point $(W^+, W^-)$, we define the set $\mathcal{P} := \{(i,j) : (\nabla h(W^+ + W^-))_{ij} > 0\}$, i.e. the set of $(i,j)$ with directed walks from $j$ to $i$, according to Lemma 2.

**Lemma 6.** *If $(W^+, W^-)$ satisfies the KKT conditions in* (8)*, then $W^* = W^+ - W^-$ satisfies the optimality conditions in* (10) *for $\mathcal{Z} = \mathcal{P}$. If in addition $F(W)$ is convex, then $W^*$ is a minimizer of* (9) *for $\mathcal{Z} = \mathcal{P}$.*

Under the assumption that $F$ is convex, we can use Lemma 6 to show that the KKT conditions (8) are *sufficient* for local minimality in (6), despite the constraint $h(|W|) \leq 0$ not being convex.

**Theorem 7.** *Assume that $F(W)$ is convex. Then if $(W^+, W^-)$ satisfies the KKT conditions in* (8)*, $W^* = W^+ - W^-$ is a local minimum for problem* (6).

In the opposite direction of Lemma 6, we focus on the case in which a minimizer $W^*(\mathcal{Z})$ of (9) is feasible, i.e. $h(A^*(\mathcal{Z})) = 0$ for $A^*(\mathcal{Z}) = |W^*(\mathcal{Z})|$. Then by Lemma 3, we must have $(W^*(\mathcal{Z}))_{ij} = 0$ wherever $\left(\nabla h(A^*(\mathcal{Z}))\right)_{ij} > 0$. If $\mathcal{Z}$ does not include such a pair $(i,j)$, we may add $(i,j)$ to $\mathcal{Z}$ while preserving the optimality of the existing solution $W^*(\mathcal{Z})$ with respect to (9) (since it already satisfies the new constraint $W_{ij} = 0$). Hence for feasible $W^*(\mathcal{Z})$, we adopt the convention that all $(i,j)$ with $(W^*(\mathcal{Z}))_{ij} = 0$ and $\left(\nabla h(A^*(\mathcal{Z}))\right)_{ij} > 0$ are included in $\mathcal{Z}$.

We call $\mathcal{Z}$ *irreducible* if it contains *only* pairs $(i,j)$ for which $\left(\nabla h(A^*(\mathcal{Z}))\right)_{ij} > 0$.

**Theorem 8.** *If a minimizer $W^*(\mathcal{Z})$ of* (9) *is feasible and $\mathcal{Z}$ is irreducible, then $W^+ = (W^*(\mathcal{Z}))_+$, $W^- = (W^*(\mathcal{Z}))_-$ satisfy the KKT conditions in* (8).

If $W^*(\mathcal{Z})$ is feasible but $\mathcal{Z}$ is not irreducible, then the following result guarantees that $\mathcal{Z}$ may be reduced to an irreducible set without losing feasibility. We assume that $F(W)$ is separable (decomposable) as $F(W) = \sum_{j=1}^{d} F_j(W_{\cdot j})$.

**Lemma 7.** *Assume that the score function $F(W)$ is separable. Suppose that $W^*(\mathcal{Z})$ in* (9) *is feasible and $\mathcal{Z}_0(j) = \{(i_1, j), \ldots, (i_J, j)\} \subseteq \mathcal{Z}$ is a subset for which $\left(\nabla h(A^*(\mathcal{Z}))\right)_{ij} = 0$, $(i,j) \in \mathcal{Z}_0(j)$. Then $W^*(\mathcal{Z} \backslash \mathcal{Z}_0(j))$ is also feasible.*

Since the removal of a constraint $(i,j) \in \mathcal{Z}$ for which $\left(\nabla h(A^*(\mathcal{Z}))\right)_{ij} = 0$ does not affect feasibility, we call such a constraint *unnecessary* as a somewhat colloquial shorthand.

The development in this subsection suggests the meta-algorithm in Algorithm 1, which we refer to as KKT-informed local search. An instantiation is described in Section 4.2.

**Theorem 9.** *If $F(W)$ is separable, KKT-informed local search satisfies the KKT conditions* (8).

When combined with Theorem 7 and a convex $F(W)$, Theorem 9 guarantees that KKT-informed local search will result in local minima. However, due to the non-convex constraint, the quality of such local minima is highly dependent on the particular instantiation of the meta-algorithm. Section 5 shows for example that the choice of initialization plays a large role.

---
**Algorithm 1** KKT-informed local search (KKTS)
---
**Require:** Initial set $\mathcal{Z}$ of edge absence constraints. Solve (9).

 1: **while** $W^*(\mathcal{Z})$ infeasible **do**
 2:      Select edge(s) in cycle $((W^*(\mathcal{Z}))_{ij} \neq 0, \left(\nabla h(A^*(\mathcal{Z}))\right)_{ij} > 0)$. Add to $\mathcal{Z}$. Re-solve (9).

 3: **while** $\mathcal{Z}$ reducible **do**
 4:      Remove one or more unnecessary constraints $(i,j) \in \mathcal{Z}$ (see Lemma 7). Re-solve (9).
---

## 4 Algorithms

For the algorithms in this section, we let the score function $F(W)$ be the sum of a smooth loss function $\ell(W; \mathbf{X})$ with respect to the data $\mathbf{X}$ and an $\ell_1$ penalty to promote overall sparsity, as in [31]:

$$F(W) = \ell(W; \mathbf{X}) + \tau \|W\|_1. \tag{11}$$

### 4.1 Augmented Lagrangian with absolute value adjacency matrix

Formulation (6) naturally suggests an augmented Lagrangian algorithm as in [31] but with $h(|W|)$ instead of $h(W \circ W)$. Using the $(W^+, W^-)$ representation as in (7), the augmented Lagrangian minimized in each iteration is

$$L(W^+, W^-, \alpha, \rho) = \ell\left(W^+ - W^-; \mathbf{X}\right) + \tau \mathbf{1}^T \left(W^+ + W^-\right)\mathbf{1} + \alpha h\left(W^+ + W^-\right) + \frac{\rho}{2} h\left(W^+ + W^-\right)^2,$$

subject to $W^+ \geq 0$ and $W^- \geq 0$, where $\mathbf{1}$ is a vector of ones. The gradients are given by

$$\nabla_{W^\pm} L(W^+, W^-, \alpha, \rho) = \pm \nabla \ell\left(W^+ - W^-; \mathbf{X}\right) + \tau \mathbf{1}\mathbf{1}^T + \left(\alpha + \rho h\left(W^+ + W^-\right)\right)\nabla h\left(W^+ + W^-\right).$$

We otherwise closely follow the algorithm in [31].

### 4.2 KKT-informed local search

We now describe an instantiation of the KKT-informed local search meta-algorithm in Algorithm 1, covering initializing the set $\mathcal{Z}$ of edge absence constraints, selecting edges for removal (line 2), reducing unnecessary constraints (line 4), and re-solving (9). We also discuss an additional operation of reversing edges, which is not part of Algorithm 1 but helps in attaining better local minima.

**Initializing $\mathcal{Z}$** Given any matrix $W$ as an initial solution, we set to zero elements in $W$ that are smaller than a threshold $\omega$ in absolute value. The set $\mathcal{Z}$ is then defined as $\mathcal{Z} = \{(i,j) : W_{ij} = 0\}$.

**Selecting edges for removal (line 2)** We consider an approach of minimizing the Lagrangian $F(W) + \alpha h(|W|)$ of (6) subject to the existing constraints $W_{ij} = 0$ for $(i,j) \in \mathcal{Z}$. For $\alpha = 0$, the minimizer is the existing solution $W^*(\mathcal{Z})$, and as $\alpha$ increases, weights $W_{ij}$ will be set to zero to decrease the infeasibility penalty $h(|W|)$, trading off against the score function $F(W)$.

We implement a computationally simple version of the above idea. First, $h(A) = h(|W|)$ in the Lagrangian is linearized around $A^*(\mathcal{Z}) = |W^*(\mathcal{Z})|$ as $h(A) \approx h\left(A^*(\mathcal{Z})\right) + \left\langle \nabla h\left(A^*(\mathcal{Z})\right), A - A^*(\mathcal{Z}) \right\rangle$. After dropping constant terms and expanding the inner product, the constrained, linearized Lagrangian to be minimized is as follows:

$$\min_W \; F(W) + \alpha \sum_{(i,j):i \neq j} \left(\nabla h(A^*(\mathcal{Z}))\right)_{ij} |W_{ij}| \quad \text{s.t.} \quad W_{ij} = 0, \quad (i,j) \in \mathcal{Z}. \tag{12}$$

Problem (12) is a score minimization problem with a weighted $\ell_1$ penalty and parameters $W_{ij}$, $(i,j) \in \mathcal{Z}$ being absent. Furthermore, if $F(W)$ is separable column-wise, (12) is also separable.

Second, we follow the solution *path* of (12), defined by $\alpha$, from $W^*(\mathcal{Z})$ at $\alpha = 0$ only until the first existing edge belonging to a cycle $((W^*(\mathcal{Z}))_{ij} \neq 0, \left(\nabla h(A^*(\mathcal{Z}))\right)_{ij} > 0)$ is set to zero. If $\ell(W; \mathbf{X})$ in (11) is the least-squares loss, the solution path is piecewise linear and we have implemented a modified version of the LARS algorithm [11] to efficiently track the path. The modification accounts for the non-uniformity of the weights $\left(\nabla h(A^*(\mathcal{Z}))\right)_{ij}$, some of which may be zero, in the $\ell_1$ penalty in (12). It is described further in the SM.

**Reducing unnecessary constraints (line 4)** We also refer to this step as restoring edges ("restore" because these edges were likely present in an earlier iteration when $W$ was denser), in analogy with the previous step which removes edges. When there are multiple unnecessary constraints, the order in which they are removed can matter because the removal of constraints and re-optimization of (9) can make previously unnecessary constraints necessary. Because of this, even though Lemma 7 allows for multiple unnecessary constraints $(i_1, j), \ldots, (i_J, j)$ to be removed at a time, we opt to do so only one at a time. Given multiple unnecessary constraints $(i, j)$, we greedily choose one for which the absolute partial derivative of the loss function, $|(\nabla \ell(W; \mathbf{X}))_{ij}|$, is largest. This strategy gives the largest instantaneous rate of decrease of the loss as the constraint $W_{ij} = 0$ is relaxed.

**Reversing edges** In addition to removing and restoring edges, we consider reversing edges, which involves two operations: adding $(i, j)$ to $\mathcal{Z}$ to remove an existing edge $(W^*(\mathcal{Z}))_{ij} \neq 0$, and removing $(j, i)$ from $\mathcal{Z}$ (which must have been a necessary constraint if $W^*(\mathcal{Z})$ is feasible, to avoid a 2-cycle) to introduce the opposite edge. In contrast to removing edges, which generally increases $F(W)$ but decreases $h(A)$, and restoring edges, which decreases $F(W)$ and is guaranteed by Lemma 7 not to increase $h(A)$, reversing edges does not necessarily decrease $F(W)$ or $h(A)$. We therefore *accept* an edge reversal only if it decreases one of $F(W)$, $h(A)$ relative to the original direction and does not increase the other, and otherwise *reject* the reversal.

There are many possible variations in when to perform edge reversals within Algorithm 1. In our implementation, we restrict reversals to the second while-loop and alternate between restoring one edge (reducing $\mathcal{Z}$ by one) and attempting all possible reversals given the current state. When there are multiple reversal candidates, similar to restoring edges, we evaluate the loss partial derivatives $|(\nabla \ell(W; \mathbf{X}))_{ji}|$, this time associated with introducing the reverse edges $(j, i)$, and proceed in order of decreasing $|(\nabla \ell(W; \mathbf{X}))_{ji}|$.

The edge reversal operation is made much more efficient by keeping a memory of previously attempted reversals that do not have to be attempted again for some time. When the reversal of edge $(i, j)$ is attempted, it is recorded in the memory, and if the reversal is accepted, reversal of $(j, i)$ is also added to the memory as it would revert to the previous inferior state. The memory for $(i, j)$ is cleared when either column $i$ or $j$ is updated since this may change the value of reversing $(i, j)$.

**Re-solving (9) (lines 2, 4)** Removing, restoring, and reversing edges all involve re-solving (9) after adding to $\mathcal{Z}$, reducing $\mathcal{Z}$, or both in the case of reversals. When $\ell(W; \mathbf{X})$ in (11) is the least-squares loss, these re-optimizations can be done efficiently using the LARS algorithm. In the case of adding $(i, j)$ to $\mathcal{Z}$, an increasing penalty is imposed on $|W_{ij}|$, while in the case of removing $(i, j)$ from $\mathcal{Z}$, a penalty equivalent to the constraint $W_{ij} = 0$ is inferred and then decreased to zero. Further details are in the SM.

# 5 Experiments

We compare the structure learning performance of the following base algorithms: NOTEARS [31], the FGS implementation [21] of GES [7], MMHC [27], PC [26], augmented Lagrangian with absolute value adjacency matrix $A = |W|$ (Section 4.1, abbreviated 'Abs'), and KKT-informed local search (Section 4.2, KKTS) initialized with the unconstrained solution ($\mathcal{Z} = \{(i, i), i \in \mathcal{V}\}$ just to avoid self-loops). We also experimented with CAM [6] but defer those results to the SM as we found them less competitive in the tested settings. In addition, we use each of the above base algorithms to initialize KKTS (denoted by appending '-KKTS' and excepting KKTS itself). Algorithm parameter settings are detailed in the SM. Of note are the default termination tolerance on $h$, $\epsilon = 10^{-10}$, and the threshold on $W$, $\omega = 0.3$ following [31], applied after NOTEARS, Abs, and KKTS as well as to initialize $\mathcal{Z}$ before KKTS.

The experimental setup is similar to [31]. In brief, random Erdös-Rényi or scale-free graphs are generated with $kd$ expected edges (denoted ER$k$ or SF$k$), and uniform random weights $W$ are assigned to the edges. Data $\mathbf{X} \in \mathbb{R}^{n \times d}$ is then generated by taking $n$ i.i.d. samples from the linear SEM $X = W^T X + z$, where $z$ is either Gaussian, Gumbel, or exponential noise. 100 trials are performed for each graph type-noise type combination, which is an order of magnitude larger than in e.g. [31, 29] and reduces the standard errors of the estimated means.

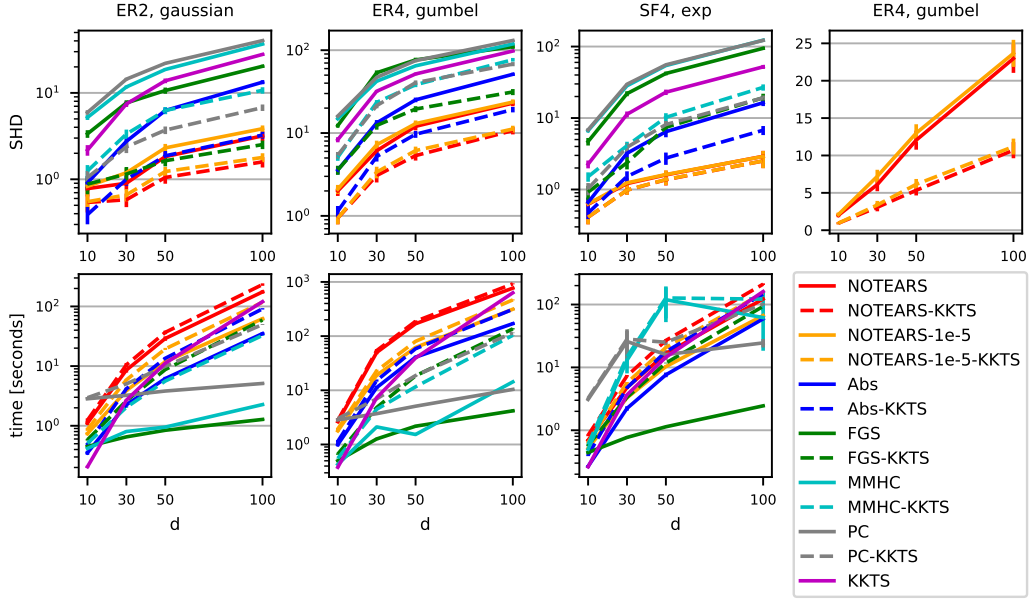

Figure 1: Structural Hamming distances (SHD) with respect to true graph and solution times for $n = 1000$. Error bars indicate standard errors over 100 trials. Red lines overlap with orange in the SF4 SHD plot. The upper right panel focuses on combinations with NOTEARS using a linear vertical scale.

Figure 1 shows structural Hamming distances (SHD) with respect to the true graph and running times for three graph-noise combinations and $n = 1000$. Figure 2 shows the same metrics and combinations for the more challenging setting $n = 2d$, with largely similar patterns. Other graph-noise combinations, results in tabular form, and computing environment details are in the SM.

We focus first on the base algorithms (solid lines), of which NOTEARS is clearly the best in terms of SHD.[1] Abs is next and better than FGS, MMHC, and PC. We hypothesize that the smoothness of the quadratic adjacency $A = W \circ W$ used by NOTEARS is better able to overcome non-convexity than the non-smooth $A = |W|$ of Abs, which tends to force parameters $W_{ij}$ to zero, perhaps too soon. The non-convexity is further reflected in the inferior performance of (pure) KKTS, which only takes local steps starting from the unconstrained solution.

We now turn to the '-KKTS' combinations (dashed lines). It is seen that KKTS, and the theoretical understanding it embodies, improve the SHD of *all* base algorithms (including CAM in the SM). The improvement is by at least a factor of 2, except when the SHD is already low (e.g. NOTEARS on SF4), and moreover is consistent across dimensions $d$. An ablation study in the SM shows that both reducing unnecessary constraints and reversing edges contribute to the improvement.

In the case of NOTEARS-KKTS, while Proposition 3 asserts that NOTEARS cannot yield an exactly feasible solution, let alone a KKT point, Figure 1 confirms that it yields high-quality nearly feasible solutions. NOTEARS is therefore well-suited as an initialization for KKTS, and combining them apparently results in new state-of-the-art accuracy. Furthermore, in an attempt to achieve feasibility, NOTEARS uses more augmented Lagrangian iterations and very large penalty parameters $\alpha$ and $\rho$. The latter causes the augmented Lagrangian (5) to be poorly conditioned and optimization solvers for it to take longer to converge. Thus, to reduce solution time as well as satisfy KKT conditions, we terminate NOTEARS early with a higher $h$ tolerance of $\epsilon = 10^{-5}$ before running KKTS. Figure 1 shows that this results in nearly the same SHD improvement over NOTEARS while also taking

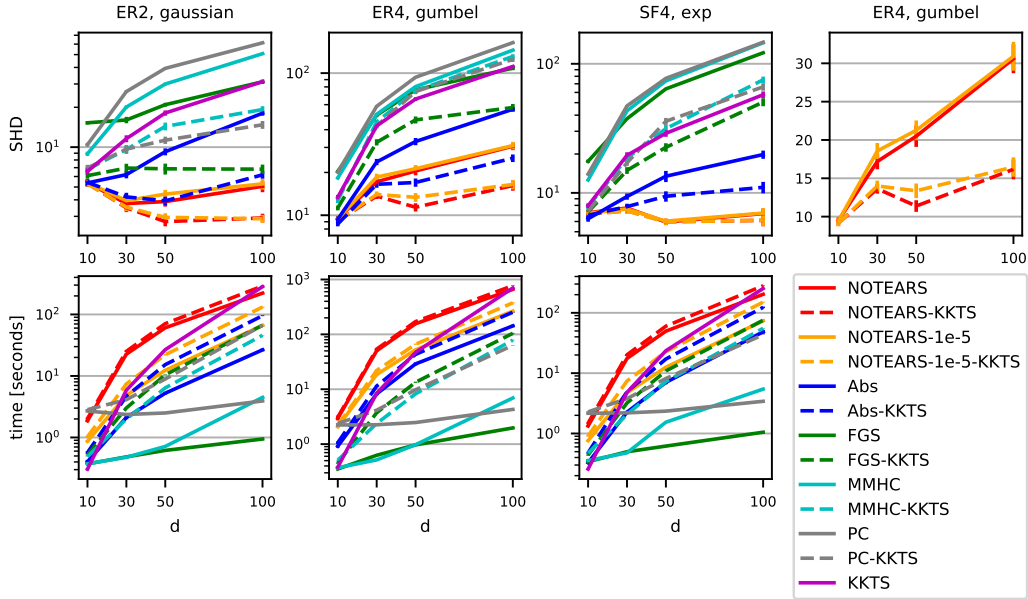

Figure 2: Structural Hamming distances (SHD) with respect to true graph and solution times for $n = 2d$. Red lines overlap with orange in the SF4 SHD plot. The upper right panel focuses on combinations with NOTEARS using a linear vertical scale.

considerably less time (except for SF4). Abs-KKTS similarly outperforms NOTEARS on ER graphs and takes even less time.

# 6 Conclusion and future work

We have re-examined a recently proposed continuous optimization framework for learning Bayesian networks. Our most important contributions are as follows: (1) better understanding of the NOTEARS formulation and algorithm of [31]; (2) analysis and understanding of the KKT optimality conditions for an equivalent reformulation (for which they do indeed hold); (3) a local search algorithm informed by the KKT conditions that significantly and universally improves the accuracy of NOTEARS and other algorithms.

A clear next step is to generalize the theory and algorithms to the case in which each edge in the graph corresponds to multiple parameters. One motivation is to allow nonlinear models; a nonlinear extension of the absolute value case of Section 3.2 could parallel the recent nonparametric extension [32] for the quadratic case. Another reason for having multiple parameters is to accommodate non-binary categorical variables, which are typically encoded into multiple binary variables on the input side, or predicted using e.g. multi-logit regression [14] on the output side. Other future directions include improving the efficiency of algorithms for solving (3), (6) and exploring alternative acyclicity characterizations from Section 2.

## Broader Impact

Bayesian networks are fundamentally about modeling the joint probability distribution of data, in a parsimonious and comprehensible manner. This work therefore contributes mostly to layer 0 ("foundational research") in the "Impact Stack" of [3], particularly with regard to the theoretical aspects. If one views Bayesian network structure learning as a "ML technique" rather than a "foundational technique", then the algorithmic contribution also falls into layer 1. We thus confine our discussion of broader impacts mostly to layers 0 and 1, i.e. "tractable" impacts according to [3], as it is difficult and perhaps inappropriate to speculate further.

The predominant contribution of this work is to theoretical understanding of the optimization problem that is score-based structure learning, and specifically a continuous formulation thereof. This understanding has resulted in improvements in accuracy (as measured by structural Hamming distance), and we expect that further improvements will be made in future work. We also believe that this understanding may lead to advances in computational efficiency as well, beyond the simple measure of terminating the NOTEARS algorithm early when it has no hope of reaching feasibility, or observing that the absolute value version (Abs) converges more quickly. For example, new optimization algorithms may be proposed for problems (3) and/or (6) that take better advantage of their properties.

As the accuracy and scalability of Bayesian network structure learning continue to increase, we hope that it becomes an even more commonly used technique for modeling data than it is now. We are particularly interested in its use as the first step in *causal* structure discovery, which may then facilitate other causal inference tasks. We recognize however that errors in structure learning may compound into potentially more serious downstream errors. This is an issue calling for further study.

## Funding Acknowledgments

Y. Yu is supported by the National Science Foundation under award DMS 1753031.

## Footnotes

[1]The SHDs for NOTEARS and FGS in Figure 1 are much better than those reported in [31], by almost an order of magnitude in some cases. Part of the improvement is due to code updates for NOTEARS but the rest we cannot explain. We also show in the SM that subtracting the mean from $\mathbf{X}$ improves the SHD by a noticeable factor for some noise types. All results in Figure 1 are obtained with zero-mean $\mathbf{X}$.

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
