[Supplementary Material]

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

# A  Proofs

## A.1  Proofs for Section 2

### A.1.1  Proof of Lemma 1

Given a weighted adjacency matrix $A$, we define the *weight* of a directed walk from $i$ to $j$ to be the product $A_{i,i_1} A_{i_1,i_2} \dots A_{i_{l-1},j}$. It is well-known that $(A^p)_{ij}$ is the sum of the weights of all length-$p$ directed walks from $i$ to $j$ [5]. Therefore $\operatorname{tr}(A^p)$ is the sum of the weights of all length-$p$ directed circuits. If $\mathcal{G}$ is acyclic, then all of these sums are zero, i.e. $A$ is nilpotent according to the definition. The converse also holds.

### A.1.2  Proof of Theorem 1

Using Lemma 1, we equivalently show that $A$ is nilpotent if and only if $h(A) = 0$. The "only if" direction is clearly true.

If $h(A) = 0$, then because $c_p > 0$, $p = 1, \dots, d$, and $\operatorname{tr}(A^p) \geq 0$ due to the non-negativity of $A$, we must have $\operatorname{tr}(A^p) = 0$, $p = 1, \dots, d$. The extension to higher powers of $A$ can be shown by induction using the Cayley-Hamilton theorem. For the base case $d + 1$, $A^{d+1}$ can be expressed as a linear combination of $A, \dots, A^d$, specifically by multiplying the characteristic polynomial of $A$ by another power of $A$. Therefore $\operatorname{tr}(A^{d+1}) = 0$. For the inductive step $p > d + 1$, $A^p$ can similarly be expressed as a linear combination of $A^{p-d}, \dots, A^{p-1}$, the traces of which are all known to be zero. We conclude that $\operatorname{tr}(A^p) = 0$ for all $p \in \mathbb{N}$.

### A.1.3  Proof of Lemma 2

From the power series expression for $\nabla h(A)$,

$$(\nabla h(A))_{ij} = \sum_{p=1}^{d} p c_p \left(A^{p-1}\right)_{ji} = \sum_{p=1}^{d-1} (p+1) c_{p+1} \left(A^p\right)_{ji} \tag{13}$$

for $i \neq j$. Thus if $(\nabla h(A))_{ij} > 0$, then $(A^p)_{ji} > 0$ for at least one $p$, i.e. there exists a directed walk of length $p$ from $j$ to $i$.

Conversely, if there is a directed walk from $j$ to $i$, then there is also a directed path from $j$ to $i$. A directed path can have length at most $d - 1$ since no vertices can be repeated. Therefore $(A^p)_{ji} > 0$ for at least one $p$ in $\{1, \dots, d-1\}$ and $(\nabla h(A))_{ij} > 0$ from (13).

### A.1.4  Proof of Lemma 3

We first make an elementary observation from the expression $\nabla h(A) = \sum_{p=1}^{d} p c_p \left(A^{p-1}\right)^T$.

**Lemma 8.** *For non-negative matrices $A$, $\nabla h(A)$ is non-negative and $h(A)$ is therefore a non-decreasing function in the sense that $h(A) \geq h(B)$ if $A - B \geq 0$.*

Lemma 3 then follows from Lemma 9 below and rewriting $\operatorname{tr}\left((\nabla h(A))^T A\right)$ as the inner product

$$\operatorname{tr}\left((\nabla h(A))^T A\right) = \sum_{i,j} (\nabla h(A))_{ij} A_{ij}.$$

Since $A$ is non-negative and $\nabla h(A)$ is also non-negative (Lemma 8), $\operatorname{tr}\left((\nabla h(A))^T A\right) = 0$ if and only if $(\nabla h(A))_{ij} A_{ij} = 0$ for all $i, j$.

**Lemma 9.** *A directed graph $\mathcal{G}$ is acyclic if and only if $\operatorname{tr}\left((\nabla h(A))^T A\right) = 0$ for any $h$ defined by* (1).

*Proof.* Again from the power series expression $\nabla h(A) = \sum_{p=1}^{d} p c_p \left(A^{p-1}\right)^T$,

$$\operatorname{tr}\left((\nabla h(A))^T A\right) = \sum_{p=1}^{d} p c_p \operatorname{tr}\left(A^p\right).$$

Similar to (1), this is a strictly positive linear combination of non-negative traces $\text{tr}(A^p)$, $p = 1, \ldots, d$. Thus $\text{tr}\left((\nabla h(A))^T A\right) = 0$ if and only if $\text{tr}(A^p) = 0$ for $p = 1, \ldots, d$. Similarly from (1), $h(A) = 0$ if and only if $\text{tr}(A^p) = 0$ for $p = 1, \ldots, d$. Theorem 1 completes the chain of equivalences. $\quad\square$

## A.2 Proofs for Section 3.1

Propositions 2 and 3 are immediate consequences of Lemma 4.

### A.2.1 Proof of Lemma 4

By the chain rule,

$$\nabla_W(h(W \circ W)) = \nabla h(W \circ W) \circ 2W,$$

where $\nabla h(W \circ W)$ refers to

$$\nabla h(A) = \sum_{p=1}^{d} pc_p \left(A^{p-1}\right)^T$$

evaluated at $A = W \circ W$. (The above gradient expression generalizes eq. (8) in [33].) If $W$ is feasible, i.e. $h(W \circ W) = 0$, then Lemma 3 with $A = W \circ W$ implies that $\nabla h(W \circ W) \circ W \circ W = 0$. Since the latter is true if and only if $\nabla h(W \circ W) \circ W = 0$, we have $\nabla_W(h(W \circ W)) = 0$.

## A.3 Equivalence of problems (6) and (7)

We map between solutions to (6) and (7) as follows:

$$W \mapsto \left(W^+, W^-\right) = \left((W)_+, (W)_-\right), \tag{14a}$$

$$\left(W^+, W^-\right) \mapsto W = W^+ - W^-, \tag{14b}$$

where

$$(W)_+ := \max\{W, 0\}, \quad (W)_- := -\min\{W, 0\},$$

and the maximum and minimum are taken element-wise. $(W)_+$ and $(W)_-$ are therefore the positive and negative parts of $W$, motivating the $W^+, W^-$ notation.

To establish the equivalence, we introduce the following intermediate formulation with the additional constraint $W^+ \circ W^- = 0$:

$$\begin{aligned} \min_{W^+, W^-} \quad & F\left(W^+ - W^-\right) \\ \text{s.t.} \quad & h\left(W^+ + W^-\right) \leq 0, \quad W^+, W^- \geq 0, \quad W^+ \circ W^- = 0. \end{aligned} \tag{15}$$

The mappings in (14) define a one-to-one correspondence between $\mathbb{R}^{d \times d}$ and non-negative pairs $(W^+, W^-)$ satisfying $W^+ \circ W^- = 0$. Thus we have the following.

**Lemma 10.** *If $W$ is a feasible solution to problem (6), then applying mapping (14a) to $W$ yields a feasible solution to (15) with the same objective value. Conversely if $(W^+, W^-)$ is a feasible solution to (15), then $W = W^+ - W^-$ is a feasible solution to (6) with the same objective value.*

*Proof.* Mapping (14a) satisfies the constraints $W^\pm \geq 0$ and $W^+ \circ W^- = 0$. Under this last condition, we also have $W^+ + W^- = |W|$. These facts show that (14) preserves feasibility in both directions. Since $(W)_+ - (W)_- = W$, (14a) preserves the objective value, and clearly (14b) does as well. $\quad\square$

We now show that the additional constraint $W^+ \circ W^- = 0$ in (15) does not change the optimal value, i.e. there is no advantage from dropping it.

**Lemma 11.** *If $(W^+, W^-)$ is a feasible solution to problem (7) and $W^+ \circ W^- \neq 0$, then there exists a feasible solution $\left(W_0^+, W_0^-\right)$ with the same objective value and satisfying $W_0^+ \circ W_0^- = 0$.*

*Proof.* For any $(i, j)$ such that $W_{ij}^+ W_{ij}^- > 0$, we can obtain another feasible solution by reducing each of $W_{ij}^+$, $W_{ij}^-$ by the same amount until $W_{ij}^+ W_{ij}^- = 0$. Since the objective is a function of $W^+ - W^-$, its value is unchanged. At the same time, Lemma 8 ensures that $h(W^+ + W^-)$ cannot increase since it is a non-decreasing function, and thus the solution remains feasible. $\quad\square$

In particular, an optimal solution to (7) not satisfying $W^+ \circ W^- = 0$ can be reduced to another optimal solution that does satisfy $W^+ \circ W^- = 0$. Hence it suffices to solve (15) in order to solve (7).

The combination of Lemmas 10 and 11 yields the following equivalence:

**Proposition 10.** *If $W^*$ is an optimal solution to problem* (6)*, then applying mapping* (14a) *to $W^*$ yields an optimal solution $(W^{+*}, W^{-*})$ to* (7)*. Conversely if $(W^{+*}, W^{-*})$ is an optimal solution to* (7)*, then $W^* = W^{+*} - W^{-*}$ is an optimal solution to* (6)*.*

### A.4 Proofs for Section 3.2.1

#### A.4.1 Proof of Proposition 4

To begin, we recall that a feasible solution to an inequality-constrained problem such as (7) is said to be *regular* if the gradients of the active (i.e. tight) constraints are linearly independent [4]. If a local minimum is regular, then the KKT conditions necessarily hold.

We first give expressions for the gradients of the constraints in (7). With $A = W^+ + W^-$, the gradient of $h(A)$ with respect to either $W^+$ or $W^-$ is given by $\nabla h(A)$ itself. Recalling that $W^\pm \geq 0$ is a collection of constraints $W_{ij}^\pm \geq 0$, the gradient of (say) constraint $W_{ij}^+ \geq 0$ is a matrix $E^{ij}$ with entry $(i, j)$ equal to 1 and 0 elsewhere. A linear combination of these gradients with respect to $W^+$ (respectively $W^-$) can be represented as a matrix $M^+$ (respectively $M^-$). It will be seen shortly that we can take a non-negative linear combination of these gradients, so $M^+$, $M^-$ are non-negative and we reuse the symbol $M$ from (8a).

If $(W^+, W^-)$ is feasible, then we must have $h(A) = 0$ so the constraint $h(A) \leq 0$ is active. Consider then the equation

$$M^+ = M^- = \nabla h(A), \tag{16}$$

which expresses the gradient of the constraint $h(A) \leq 0$ (with respect to $W^+$ or $W^-$) as a linear combination of gradients of the constraints $W_{ij}^+ \geq 0$ or $W_{ij}^- \geq 0$. More specifically, $M^+$ and $M^-$ in (16) are linear combinations only of those gradients $(i, j)$ for which $(\nabla h(A))_{ij} > 0$. By Lemma 3, $h(A) = 0$ implies that

$$\nabla h(A) \circ A = \nabla h(A) \circ (W^+ + W^-) = 0.$$

In particular, if $(\nabla h(A))_{ij} > 0$, then $W_{ij}^+ = W_{ij}^- = 0$, i.e. these two constraints are active. Thus $M^+$, $M^-$ are linear combinations of active constraint gradients only, and (16) equates these linear combinations to the gradient of active constraint $h(A) \leq 0$. We conclude that $(W^+, W^-)$ is not regular.

#### A.4.2 Proof of Proposition 5

Quasinormality is a weaker constraint qualification than regularity and is described in [4, Sec. 3.3.5, p. 336]. We follow the framework therein. We let the convex set $\mathcal{X}$ be $\mathbb{R}_+^{d \times d} \times \mathbb{R}_+^{d \times d}$, the set of pairs of non-negative matrices, to account for the constraints $W^\pm \geq 0$. Thus $h(A) \leq 0$ remains as a single inequality constraint, where again $A = W^+ - W^-$.

A feasible solution $(W^+, W^-)$ is *not* quasinormal if it satisfies conditions (i)–(iv) in [4, Sec. 3.3.5, p. 336]. Translated to the current case of a single inequality constraint, these conditions are (i)

$$\sum_{i,j} (\nabla h(A))_{ij} \left( W_{ij}'^+ + W_{ij}'^- - W_{ij}^+ - W_{ij}^- \right) \geq 0 \quad \forall \, (W'^+, W'^-) \in \mathcal{X}, \tag{17}$$

and (iv) in every neighborhood around $(W^+, W^-)$ (e.g. $\ell_2$ balls), there exists a $(W'^+, W'^-) \in \mathcal{X}$ for which $h(W'^+ + W'^-) > 0$. Conditions (ii) and (iii) are easily satisfied by setting the single multiplier $\mu = 1$.

To show that condition (i) (17) is satisfied, we consider the cases $(\nabla h(A))_{ij} > 0$ and $(\nabla h(A))_{ij} = 0$. In the former case, since $(W^+, W^-)$ is feasible, Lemma 3 requires that $A_{ij} = W_{ij}^+ + W_{ij}^- = 0$. Hence the corresponding term in (17) becomes $(\nabla h(A))_{ij} \left( W_{ij}'^+ + W_{ij}'^- \right)$ and is always non-negative. In the latter case $(\nabla h(A))_{ij} = 0$, the contribution to the sum is zero. Therefore (17) is satisfied.

Condition (iv) can be satisfied by choosing $W' = W'^+ - W'^-$ to be a fully dense matrix (corresponding to a complete graph) that is arbitrarily close to $W = W^+ - W^-$. Concretely, let $W'^- = W^-$,

$W_{ij}'^+ = \epsilon$ wherever $W_{ij}^+ = W_{ij}^- = 0$, and $W_{ij}'^+ = W_{ij}^+$ otherwise. Then $h(W'^+ + W'^-) > 0$ for all $\epsilon > 0$.

### A.4.3 Proof of Lemma 5

We provide a graphical proof by viewing $A$ as an adjacency matrix and $(\nabla h(A))_{ij} > 0$ as an indicator of a directed walk from node $j$ to $i$, the latter as ensured by Lemma 2. If $(\nabla h(A))_{ij} > 0$, i.e. there exists a directed walk from $j$ to $i$, then there also exists a directed path from $j$ to $i$. Since a directed path connects distinct vertices, it cannot contain an edge $(k, j)$. (Any directed walk from $j$ to $i$ that does have an edge $(k, j)$ must have a final subwalk from $j$ to $i$ that is a path.) Thus changing the values of $A_{kj}$, and specifically removing edges into $j$, cannot remove directed paths from $j$ to $i$ (and thereby set $(\nabla h(A))_{ij} = 0$).

Similarly for the second statement, if $(\nabla h(A))_{ij} = 0$, then there is no directed walk from $j$ to $i$, including directed paths. Then changing the values of $A_{kj}$, and specifically adding edges into $j$, cannot create a directed walk from $j$ to $i$ because it would require a final subwalk from $j$ to $i$ that is a directed path, which was assumed not to exist.

### A.4.4 Proof of Theorem 6

By definition, $(W^+, W^-)$ is a feasible solution to (7). We prove that (8a) and (8b) can be satisfied. Again letting $A = W^+ + W^-$, we consider two cases for the entries of the constraint gradient $\nabla h(A)$.

**Case $(\nabla h(A))_{ij} = 0$:** In this case, the only way in which (8a) can be satisfied is if $(\nabla F(W^+ - W^-))_{ij} = 0$, and we show that this is indeed true. First we establish by a graphical argument that all $(\tilde{W}^+, \tilde{W}^-)$ of the form $\tilde{W}^+ = W^+ + wE^{ij}$ and $\tilde{W}^- = W^-$ are feasible solutions to (7), where $W_{ij}^+ + w \geq 0$ to maintain non-negativity. The only potential obstacle is if $W_{ij}^+ = W_{ij}^- = 0$ so that varying $w$ introduces an edge $(i, j)$. However, since $(\nabla h(A))_{ij} = 0$, there is no directed walk from $j$ to $i$, and Lemma 5 ensures that none can be created by varying $\tilde{W}_{ij}^+$. Therefore $(\tilde{W}^+, \tilde{W}^-)$ remains acyclic and feasible. The above argument can be repeated for $\tilde{W}^+ = W^+$ and $\tilde{W}^- = W^- + wE^{ij}$.

From the previous paragraph, we conclude that $W = W^+ - W^- + wE^{ij}$ is feasible for all $w \in \mathbb{R}$. Then if $(W^+, W^-)$ is a local minimum, we must have the partial derivative $(\nabla F(W^+ - W^-))_{ij} = 0$. Otherwise, entry $(i, j)$ could be increased or decreased ($w > 0$ or $w < 0$) to reduce the cost while remaining feasible.

Given that $(\nabla h(A))_{ij} = (\nabla F(W^+ - W^-))_{ij} = 0$, we take $M_{ij}^+ = M_{ij}^- = 0$ to satisfy component $(i, j)$ of constraint (8b) as well as (8a).

**Case $(\nabla h(A))_{ij} > 0$:** Since $(W^+, W^-)$ is feasible, $h(A) = 0$ and Lemma 3 implies that

$$(\nabla h(A))_{ij} A_{ij} = (\nabla h(A))_{ij} (W_{ij}^+ + W_{ij}^-) = 0.$$

Hence $W_{ij}^+ = W_{ij}^- = 0$, satisfying (8b).

To satisfy (8a), we take

$$\lambda \geq \max_{(i,j):(\nabla h(A))_{ij}>0} \frac{\left| (\nabla F(W^+ - W^-))_{ij} \right|}{(\nabla h(A))_{ij}}$$

and define $M_{ij}^+$, $M_{ij}^-$ to be the resulting slack in component $(i, j)$ of (8a). This completes the proof.

The above proof is related to the idea discussed in [4] that the directions of first-order feasible variations around $(W^+, W^-)$ do not include a direction of descent. The latter idea is used to prove existence of Lagrange multipliers in [4, Prop. 3.3.14].

## A.5 Proofs for Section 3.2.2

### A.5.1 Proof of Lemma 6

The proof follows from that of Theorem 6. Case $(\nabla h(A))_{ij} = 0$ in Theorem 6 corresponds to $(i, j) \notin \mathcal{P}$ and was shown to imply $(\nabla F(W^+ - W^-))_{ij} = (\nabla F(W^*))_{ij} = 0$. Case $(\nabla h(A))_{ij} > 0$

corresponds to $(i, j) \in \mathcal{P}$ and implies $W_{ij}^+ = W_{ij}^- = W_{ij}^* = 0$. If $F$ is convex, then conditions (10) are also sufficient for optimality in (9).

### A.5.2 Proof of Theorem 7

Let $W$ be a feasible solution to (6) with $\|W - W^*\|_F < \epsilon$ (the Frobenius norm is used for concreteness), $A = |W|$, and $A^* = |W^*|$. Since the gradient $\nabla h(A) = \sum_{p=1}^{d} p c_p \left( A^{p-1} \right)^T$ is a continuous function of $A$ and therefore of $W$, there exists a sufficiently small $\epsilon > 0$ such that $(\nabla h(A))_{ij} > 0$ wherever $(\nabla h(A^*))_{ij} > 0$, in other words for $(i, j)$ in the set $\mathcal{P}$. Then for feasible $W$ within such an $\epsilon$-ball around $W^*$, it follows from Lemma 3 that $A_{ij} = W_{ij} = 0$ for $(i, j) \in \mathcal{P}$. $W$ is therefore a feasible solution to (9) for $\mathcal{Z} = \mathcal{P}$. By Lemma 6 and the convexity of $F$, we then have $F(W^*) \leq F(W)$ for all feasible $W$ such that $\|W - W^*\|_F < \epsilon$.

### A.5.3 Proof of Theorem 8

By assumption, $W^*(\mathcal{Z})$ is feasible. For $(i, j) \in \mathcal{Z}$, the constraint $W_{ij} = 0$ satisfies (8b). Since $\mathcal{Z}$ is irreducible, $(\nabla h(A^*(\mathcal{Z})))_{ij} > 0$. We may then choose $\lambda$ large enough as in the proof of Theorem 6 to satisfy (8a).

For $(i, j) \notin \mathcal{Z}$, the optimality conditions (10) imply $(\nabla F(W^*(\mathcal{Z})))_{ij} = 0$. If $(W^*(\mathcal{Z}))_{ij} \neq 0$, then we must have $(\nabla h(A^*(\mathcal{Z})))_{ij} = 0$ by the feasibility of $W^*(\mathcal{Z})$ and Lemma 3. If $(W^*(\mathcal{Z}))_{ij} = 0$, then the convention in defining $\mathcal{Z}$ also ensures that $(\nabla h(A^*(\mathcal{Z})))_{ij} = 0$. Letting $M_{ij}^+ = M_{ij}^- = 0$ then satisfies (8a) and (8b).

### A.5.4 Proof of Lemma 7

Since $F(W)$ is separable and the pairs in $\mathcal{Z}_0(j)$ have $j$ in common, removing the constraints $W_{ij} = 0$ for $(i, j) \in \mathcal{Z}_0(j)$ affects only the subproblem of (9) for node $j$. This subproblem is now given by

$$\underset{W_{\cdot j}}{\arg\min} \ F_j(W_{\cdot j}) \quad \text{s.t.} \quad W_{ij} = 0, \quad (i, j) \in \mathcal{Z} \backslash \mathcal{Z}_0(j). \tag{18}$$

By the definitions of $\mathcal{Z}$ and $\mathcal{Z}_0(j)$, we have $(\nabla h(A^*(\mathcal{Z})))_{ij} = 0$ for $(i, j) \notin \mathcal{Z} \backslash \mathcal{Z}_0(j)$, i.e. there are no directed walks from $j$ to such $i$. From Lemma 5, it follows that re-optimizing the values of $W_{ij}, (i, j) \notin \mathcal{Z} \backslash \mathcal{Z}_0(j)$ in (18) cannot create directed walks from $j$ to $i$. For $(i, j) \in \mathcal{Z} \backslash \mathcal{Z}_0(j)$, $W_{ij}$ is constrained to zero. We conclude that re-solving (18) does not introduce new cycles.

### A.5.5 Proof of Theorem 9

The first while-loop adds more and more elements to $\mathcal{Z}$, i.e. constrains more and more edges to be absent, and is hence guaranteed to eventually produce a feasible (acyclic) solution $W^*(\mathcal{Z})$. If the resulting set $\mathcal{Z}$ is not irreducible, then repeated application of Lemma 7 in the second while-loop will make it so while maintaining feasibility. The algorithm thus yields a solution satisfying the conditions of Theorem 8.

## B  Modified LARS algorithms

### B.1  Adding zero-value constraints

This appendix describes a modification of the LARS algorithm [11] to efficiently re-solve problem (9) under the following conditions: a) the score function $F(W)$ is given by (11), b) the loss function $\ell(W; \mathbf{X})$ is the least-squares loss, $\ell(W; \mathbf{X}) = \frac{1}{2n} \|\mathbf{X} - \mathbf{X}W\|_F^2$, and c) we have an optimal solution $W^*(\mathcal{Z})$ for the existing set $\mathcal{Z}$ of zero-value constraints and a new pair $(i_0, j)$ is being added to $\mathcal{Z}$.

Given conditions a) and b), $F(W)$ is separable column-wise and hence we only have to re-solve the subproblem of (9) for column $j$. Define $\mathcal{Z}^c(j) = \{i : (i, j) \notin \mathcal{Z}\}$ to be the set of rows in column $j$ that are not constrained to zero by $\mathcal{Z}$. Then the subproblem for column $j$ can be written as

$$\min_{W_{\mathcal{Z}^c(j), j}} \ \frac{1}{2n} \left\| \mathbf{X}_{\cdot j} - \mathbf{X}_{\cdot \mathcal{Z}^c(j)} W_{\mathcal{Z}^c(j), j} \right\|_2^2 + \tau \left\| W_{\mathcal{Z}^c(j), j} \right\|_1,$$

to which we wish to add the constraint $W_{i_0 j} = 0$. To simplify notation, let $y = \mathbf{X}_{\cdot j}$, $\tilde{\mathbf{X}} = \mathbf{X}_{\cdot \mathcal{Z}^c(j)}$, and $w = W_{\mathcal{Z}^c(j),j}$. Our approach is to add a penalty $\alpha |w_{i_0}|$ to the objective function, giving

$$\min_w \quad \frac{1}{2n} \|y - \tilde{\mathbf{X}} w\|_2^2 + \tau \|w\|_1 + \alpha |w_{i_0}|, \tag{19}$$

and increase $\alpha$ from zero until we obtain $w_{i_0} = 0$.

LARS is an active-set algorithm, where the active set $\mathcal{A}$ corresponds to the set of non-zero $w_i$, i.e. $\mathcal{A} = \{i : w_i \neq 0\}$. The initial active set is given by the existing optimal solution $W_{\mathcal{Z}^c(j),j}^*(\mathcal{Z})$. We assume that it includes $i_0$, as otherwise $w_{i_0} = 0$ and we are done.

In each iteration of LARS, the active elements of $w$ are updated as

$$w_{\mathcal{A}} \leftarrow w_{\mathcal{A}} - \gamma d, \tag{20}$$

where $\gamma$ is the step size and $d$ is an $|\mathcal{A}|$-dimensional direction vector determined below. The step size $\gamma$ will be made equal to the increase in $\alpha$ and is chosen to be the largest possible before a change in the active set occurs.

One set of conditions on $\gamma$ and $d$ comes from maintaining the optimality of $w$. Define

$$g = \frac{1}{n}\tilde{\mathbf{X}}^T \left(y - \tilde{\mathbf{X}} w\right) = \frac{1}{n}\tilde{\mathbf{X}}^T \left(y - \tilde{\mathbf{X}}_{\cdot \mathcal{A}} w_{\mathcal{A}}\right) \tag{21}$$

to be the negative gradient of the least-squares term in (19), where the second equality is due to $w_i$ being zero for $i \notin \mathcal{A}$. The update equation for $w$ (20) implies that the gradient changes as

$$g \leftarrow g + \gamma c, \tag{22}$$

where

$$c = \frac{1}{n}\tilde{\mathbf{X}}^T \tilde{\mathbf{X}}_{\cdot \mathcal{A}} d. \tag{23}$$

The optimality conditions of (19) for $i \in \mathcal{A}$ require

$$g_i + \gamma c_i = \begin{cases} \operatorname{sign}(w_i)\tau, & i \in \mathcal{A},\ i \neq i_0, \\ \operatorname{sign}(w_i)(\tau + \alpha + \gamma), & i = i_0, \end{cases} \tag{24}$$

where $\alpha$ is increased by $\gamma$ as mentioned. Defining $e^{i_0}$ to be the $|\mathcal{A}|$-dimensional standard basis vector with $e_{i_0}^{i_0} = 1$ and $e_i^{i_0} = 0$ otherwise, we must have $c_{\mathcal{A}} = \operatorname{sign}(w_{i_0})e^{i_0}$ from (24). This in combination with (23) determines the direction $d$:

$$d = \left(\frac{1}{n}\tilde{\mathbf{X}}_{\mathcal{A}}^T \tilde{\mathbf{X}}_{\cdot \mathcal{A}}\right)^{-1} \operatorname{sign}(w_{i_0})e^{i_0}. \tag{25}$$

To determine the step size $\gamma$, we consider the optimality conditions for $i \notin \mathcal{A}$, namely $|g_i + \gamma c_i| \leq \tau$. By expanding the absolute value function and disregarding one of the cases because it is always satisfied, we obtain

$$\gamma \leq \frac{\tau - \operatorname{sign}(c_i)g_i}{|c_i|}, \quad i \notin \mathcal{A}. \tag{26}$$

We also have the constraints $w_i - \gamma d_i \neq 0$ to maintain the current active set, which imply

$$\gamma \leq \frac{w_i}{d_i}, \quad i \in \mathcal{A}: \frac{w_i}{d_i} > 0, \tag{27}$$

where the constraint is never binding if $w_i/d_i < 0$. Combining (26) and (27) yields

$$\gamma = \min\left\{ \min_{i \in \mathcal{A}: w_i/d_i > 0} \frac{w_i}{d_i}, \ \min_{i \notin \mathcal{A}} \frac{\tau - \operatorname{sign}(c_i)g_i}{|c_i|} \right\}. \tag{28}$$

Let $i^*$ denote the minimizing index in (28). The active set is updated as

$$\mathcal{A} \leftarrow \begin{cases} \mathcal{A}\backslash\{i^*\}, & i^* \in \mathcal{A}, \\ \mathcal{A} \cup \{i^*\}, & i^* \notin \mathcal{A}. \end{cases} \tag{29}$$

Equations (20), (22), (23), (25), (28), and (29) define one iteration of the LARS algorithm. The algorithm terminates with $w_{i_0} = 0$ when $i_0$ leaves the active set.

## B.2 Relaxing zero-value constraints

We now discuss the use of the LARS algorithm to re-solve problem (9) after a pair $(i_0, j)$ is removed from the set $\mathcal{Z}$ of zero-value constraints. Other assumptions remain as in Appendix B.1, and thus we again only have to re-solve the subproblem of (9) for column $j$. Recalling the definition of $\mathcal{Z}^c(j)$ from Appendix B.1 and defining $\tilde{\mathcal{Z}}^c(j) = \mathcal{Z}^c(j) \cup \{i_0\}$, the subproblem for column $j$ can be expressed as

$$\min_{W_{\tilde{\mathcal{Z}}^c(j),j}} \frac{1}{2n} \left\| \mathbf{X}_{\cdot j} - \mathbf{X}_{\cdot \tilde{\mathcal{Z}}^c(j)} W_{\tilde{\mathcal{Z}}^c(j),j} \right\|_2^2 + \tau \left\| W_{\tilde{\mathcal{Z}}^c(j)j} \right\|_1 \quad \text{s.t.} \quad W_{i_0 j} = 0, \tag{30}$$

where we wish to relax the constraint $W_{i_0 j} = 0$.

To simplify notation as before, let $y = \mathbf{X}_{\cdot j}$, $\tilde{\mathbf{X}} = \mathbf{X}_{\cdot \tilde{\mathcal{Z}}^c(j)}$, and $w = W_{\tilde{\mathcal{Z}}^c(j),j}$. We show that problem (30) is equivalent to (19) for a sufficiently large penalty $\alpha$. Let $w^* = W^*_{\tilde{\mathcal{Z}}^c(j),j}(\mathcal{Z})$ denote the existing optimal solution of subproblem $j$, and $g^*$ be the corresponding negative loss gradient from (21). Then the optimality conditions for (19) imply that $w_{i_0} = 0$ if the loss gradient satisfies $|g^*_{i_0}| < \tau + \alpha$. Therefore $\alpha = |g^*_{i_0}| - \tau$ is the first value at which $w_{i_0}$ becomes active. If $|g^*_{i_0}| - \tau$ is non-positive, i.e. $|g^*_{i_0}| \leq \tau$, then relaxing the constraint $w_{i_0} = 0$ does not change $w_{i_0}$ as $w^*$ is still optimal without the constraint. In this case, we are done. Assuming therefore that $|g^*_{i_0}| - \tau > 0$, we initialize $\alpha = |g^*_{i_0}| - \tau$ and seek to decrease $\alpha$ to zero.

Given this initial value for $\alpha$, the modified LARS algorithm proceeds in the reverse direction of that in Appendix B.1. In each iteration, we update

$$w_{\mathcal{A}} \leftarrow w_{\mathcal{A}} + \gamma d, \tag{31}$$

$$g \leftarrow g - \gamma c, \tag{32}$$

$$\alpha \leftarrow \alpha - \gamma, \tag{33}$$

where $d$ and $c$ are still given by (25) and (23), except that when $w_{i_0}$ is still zero, we use $\text{sign}(g^*_{i_0})$ in place of $\text{sign}(w_{i_0})$ in (25) ([11] shows that these two signs must agree). The determination of the step size $\gamma$ is slightly modified from that in (28) because of the change in signs in (31), (32) relative to (20), (22):

$$\gamma = \min \left\{ \min_{i \in \mathcal{A}: w_i/d_i < 0} -\frac{w_i}{d_i}, \ \min_{i \notin \mathcal{A}} \frac{\tau + \text{sign}(c_i)g_i}{|c_i|} \right\}. \tag{34}$$

The update for the active set $\mathcal{A}$ remains as in (29).

In summary, each LARS iteration is defined by (31)–(34), (25), and (23). As mentioned, the algorithm terminates when $\alpha$ decreases to zero.

## B.3 Solution path of (12)

The LARS algorithm can also be adapted to compute the solution path of problem (12) as the penalty parameter $\alpha$ increases from zero. This adaptation differs from the one in Appendix B.1 in two respects: First, (12) involves updates to the entire matrix $W$, with a common step size $\gamma$, and not just to a single column. At the same time, assumptions a) and b) in Appendix B.1 remain in effect, allowing the computation of update directions to be done in a separable manner. Second, (12) includes a weighted $\ell_1$ penalty with weight matrix $\nabla h(A^*(\mathcal{Z}))$ instead of an unweighted $\ell_1$ penalty plus an additional penalty on a single element $w_{i_0}$. To ease notation, let $P = \nabla h(A^*(\mathcal{Z}))$.

As in Appendix B.1, in each iteration, $\alpha$ is increased by $\gamma$,

$$\alpha \leftarrow \alpha + \gamma, \tag{35}$$

and other quantities are updated accordingly. Equations (20) and (22) are generalized to matrices as follows:

$$W_{\mathcal{A}} \leftarrow W_{\mathcal{A}} - \gamma D_{\mathcal{A}}, \tag{36}$$

$$G \leftarrow G + \gamma C, \tag{37}$$

where the active set $\mathcal{A} = \{(i,j) : W_{ij} \neq 0\}$ is now a set of pairs, $D_{ij} = 0$ for $(i,j) \notin \mathcal{A}$, and

$$G = \frac{1}{n} \mathbf{X}^T (\mathbf{X} - \mathbf{X} W) \tag{38}$$

is the negative loss gradient matrix. From (36)–(38), it can be seen that

$$C = \frac{1}{n}\mathbf{X}^T\mathbf{X}D. \tag{39}$$

To determine $D_{ij}$ for $(i,j) \in \mathcal{A}$, we use the corresponding optimality conditions for (12):

$$G_{ij} + \gamma C_{ij} = \text{sign}(W_{ij})\left(\tau + (\alpha + \gamma)P_{ij}\right), \quad (i,j) \in \mathcal{A}. \tag{40}$$

Define $\mathcal{A}(j)$ to be the set of active elements in column $j$. By combining (39) and (40) and considering each column $j$ separately, we obtain

$$\text{sign}\left(W_{\mathcal{A}(j),j}\right) \circ P_{\mathcal{A}(j),j} = C_{\mathcal{A}(j),j} = \frac{1}{n}\mathbf{X}^T_{\cdot\mathcal{A}(j)}\mathbf{X}D_{\cdot j} = \frac{1}{n}\mathbf{X}^T_{\cdot\mathcal{A}(j)}\mathbf{X}_{\cdot\mathcal{A}(j)}D_{\mathcal{A}(j),j},$$

where the last inequality follows because $D_{ij} = 0$, $(i,j) \notin \mathcal{A}(j)$. Hence

$$D_{\mathcal{A}(j),j} = \left(\frac{1}{n}\mathbf{X}^T_{\cdot\mathcal{A}(j)}\mathbf{X}_{\cdot\mathcal{A}(j)}\right)^{-1}\left(\text{sign}\left(W_{\mathcal{A}(j),j}\right) \circ P_{\mathcal{A}(j),j}\right), \quad j \in \mathcal{V}. \tag{41}$$

To determine the step size $\gamma$, we consider the optimality conditions for $(i,j) \in \mathcal{Z}^c\backslash\mathcal{A}$, i.e.

$$\left|G_{ij} + \gamma C_{ij}\right| \leq \tau + (\alpha + \gamma)P_{ij}.$$

Similar to Appendix B.1, this can be reduced to the following upper bound on $\gamma$:

$$\gamma \leq \frac{\tau + \alpha P_{ij} - \text{sign}(C_{ij})G_{ij}}{|C_{ij}| - P_{ij}}, \quad (i,j) \in \mathcal{Z}^c\backslash\mathcal{A} : |C_{ij}| > P_{ij}, \tag{42}$$

whereas no bound is imposed if $|C_{ij}| \leq P_{ij}$. We also have the conditions $W_{ij} - \gamma D_{ij} \neq 0$ for $(i,j) \in \mathcal{A}$. Define $\Gamma$ as the resulting matrix of upper bounds,

$$\Gamma_{ij} = \begin{cases} \dfrac{W_{ij}}{D_{ij}}, & (i,j) \in \mathcal{A} : \dfrac{W_{ij}}{D_{ij}} > 0, \\ \dfrac{\tau + \alpha P_{ij} - \text{sign}(C_{ij})G_{ij}}{|C_{ij}| - P_{ij}}, & (i,j) \in \mathcal{Z}^c\backslash\mathcal{A} : |C_{ij}| > P_{ij}, \\ +\infty & \text{otherwise.} \end{cases} \tag{43}$$

Then we have

$$\gamma = \min_{i,j} \Gamma_{i,j}, \tag{44}$$

and given the minimizing pair $(i^*, j^*)$ from (44), we update the active set as

$$\mathcal{A} \leftarrow \begin{cases} \mathcal{A}\backslash\{(i^*,j^*)\}, & (i^*,j^*) \in \mathcal{A}, \\ \mathcal{A} \cup \{(i^*,j^*)\}, & (i^*,j^*) \in \mathcal{Z}^c\backslash\mathcal{A}. \end{cases} \tag{45}$$

The update to the active set (45) affects only $\mathcal{A}(j^*)$ in column $j^*$. We may take advantage of this by updating only column $j^*$ of $D$ and $C$, i.e. computing (41) for $j = j^*$ and (39) only for column $j^*$. The other columns are unchanged. Similarly, the upper bounds $\Gamma_{ij}$ are recomputed using (43) only for $j = j^*$. For columns other than $j^*$, it suffices to subtract the previous step size:

$$\Gamma_{ij} \leftarrow \Gamma_{ij} - \gamma, \quad j \neq j^*. \tag{46}$$

In summary, each iteration of the modified LARS algorithm is given by (35)–(37), (41), (39), (43)–(46), together with the simplification noted in the previous paragraph. The algorithm terminates as soon as $(i^*, j^*)$ coincides with an edge belonging to a cycle in the existing optimal solution $W^*(\mathcal{Z})$, i.e. $(i^*, j^*)$ such that $W^*_{i^*j^*}(\mathcal{Z}) \neq 0$ and $P_{i^*j^*} > 0$.

Table 1: Algorithm parameter settings

| parameter | symbol | value | applicable to |
|---|---|---|---|
| threshold on $W$ | $\omega$ [33] | 0.3 | NOTEARS, Abs, KKTS before and after |
| loss function | $\ell(W; \mathbf{X})$ | $\frac{1}{2n}\|\mathbf{X} - \mathbf{X}W\|_F^2$ | NOTEARS, Abs, KKTS |
| $\ell_1$ penalty parameter | $\tau$ | 0.1 | NOTEARS, Abs, KKTS |
| acyclity penalty | $h(A)$ | $\mathrm{tr}\big((I + A/d)^d\big) - d$ | NOTEARS, Abs, KKTS |
| $h$ tolerance | $\epsilon$ [33] | $10^{-10}$ | NOTEARS, Abs, KKTS |
| $h$ progress rate | $c$ [33] | 0.25 | NOTEARS, Abs |
| initial solution | $W_0$ [33] | 0 | NOTEARS, Abs |
| initial Lagrange multiplier | $\alpha_0$ [33] | 0 | NOTEARS, Abs |
| $\rho$ increase factor | | 10 | NOTEARS, Abs |
| $\rho$ maximum | | $10^{16}$ | NOTEARS, Abs |
| variablesel | | True | CAM [6] |
| pruning | | True | CAM [6] |

## C  Additional experimental details and results

### C.1  Algorithm parameter settings

Parameter settings for all algorithms are shown in Table 1. We use the least-squares loss $\ell(W; \mathbf{X}) = \frac{1}{2n}\|\mathbf{X} - \mathbf{X}W\|_F^2$ regardless of the noise type. We found the polynomial acyclicity penalty $h(A) = \mathrm{tr}\big((I + A/d)^d\big) - d$ from [31] to take less time and perform slightly better than the exponential penalty $h(A) = \mathrm{tr}\big(e^A\big) - d$ from [33] (polynomial is now also the default in the NOTEARS code). Similarly, we preferred a tolerance on $h$ of $\epsilon = 10^{-10}$ compared to $\epsilon = 10^{-8}$ in [33]. We did not attempt to tune other parameters.

For baseline method causal additive models (CAM), we use Causal Discovery Toolbox (CDT) [17] in Python and only tuned two input parameters, "variablesel" and "pruning". We found with both turned on, the results are the best.

For baseline method fast greedy equivalent search (FGS), we use py-causal package[2] in Python from Carnegie Mellon University. We use the default parameter settings and did not tune any.

For PC, we also used CDT, and for MMHC, we used the bnlearn package [25] in R by adapting CDT's interface for calling other bnlearn algorithms. The main parameter for both PC and MMHC is the significance level $\alpha$ for the conditional independence tests that they conduct. While we considered the same range of $\alpha$ values as in [2], we found $\alpha = 0.01$ or $\alpha = 0.05$ to be the best in all cases. The differences between $\alpha = 0.01$ and $\alpha = 0.05$ are not large, and in any case, PC and MMHC are not the most competitive algorithms in our experiments.

### C.2  Computing environment

Solution times were obtained using a single 2.0 GHz core of a server with 64 GB of memory (only a small fraction of which was used) running Ubuntu 16.04 (64-bit). The limitation to a single core was done to control for different multi-threading behavior of different algorithms and for different dimensions $d$.

### C.3  Effect of mean subtraction

We show the effect of subtracting the mean from the data $\mathbf{X}$ as a preprocessing step in Figure 3. Tables 2 and 3 present the same results in tabular form. As one may see, subtracting the mean improves the SHD in the ER4 Gumbel case for all the methods shown and slightly decreases the running time. Mean subtraction has less effect in the Gaussian case. In our experience, subtracting the mean improves results or at least does not hurt in all the cases we studied, not just the ones shown in Figure 3.

Figure 3: Effect of mean subtraction ('nzm' means nonzero mean) on SHD and solution times.

Table 2: Effect of mean subtraction ('nzm' means nonzero mean) on ER4 graphs with Gaussian noise

| | $d = 10$ | | | $d = 30$ | | |
|---|---|---|---|---|---|---|
| | SHD | nnz | time (sec) | SHD | nnz | time (sec) |
| NOTEARS-nzm | 3.70±0.36 | 18.05±0.29 | 2.7±0.2 | 7.66±0.81 | 58.11±0.76 | 47.9±3.5 |
| NOTEARS | 3.61±0.36 | 18.08±0.29 | 2.8±0.2 | 7.42±0.81 | 57.97±0.74 | 45.9±3.4 |
| NOTEARS-KKTS-nzm | 2.01±0.22 | 18.32±0.30 | 2.8±0.2 | 4.48±0.54 | 58.27±0.72 | 51.6±3.5 |
| NOTEARS-KKTS | 1.87±0.20 | 18.37±0.30 | 2.9±0.2 | 4.70±0.58 | 58.18±0.72 | 49.5±3.4 |
| Abs-nzm | 4.31±0.36 | 18.31±0.30 | 1.0±0.1 | 14.64±1.10 | 60.18±0.80 | 9.5±0.7 |
| Abs | 4.52±0.41 | 18.16±0.30 | 0.9±0.1 | 15.06±1.06 | 60.39±0.81 | 9.8±0.6 |
| Abs-KKTS-nzm | 2.29±0.25 | 18.29±0.31 | 1.1±0.1 | 6.13±0.64 | 58.31±0.71 | 13.0±0.7 |
| Abs-KKTS | 2.54±0.27 | 18.19±0.31 | 1.1±0.1 | 6.14±0.56 | 58.23±0.72 | 13.4±0.6 |
| FGS-nzm | 13.48±0.74 | 28.44±0.81 | 0.5±0.0 | 53.21±3.30 | 118.38±4.48 | 1.3±0.1 |
| FGS | 13.48±0.74 | 28.44±0.81 | 0.5±0.0 | 53.21±3.30 | 118.38±4.48 | 1.3±0.1 |
| FGS-KKTS-nzm | 5.26±0.55 | 17.57±0.30 | 0.7±0.0 | 15.50±1.34 | 59.73±0.86 | 4.4±0.1 |
| FGS-KKTS | 4.92±0.47 | 17.79±0.30 | 0.7±0.0 | 15.03±1.38 | 59.72±0.87 | 4.5±0.1 |
| | $d = 50$ | | | $d = 100$ | | |
| | SHD | nnz | time (sec) | SHD | nnz | time (sec) |
| NOTEARS-nzm | 12.41±1.13 | 99.59±1.06 | 157.5±7.7 | 22.61±1.73 | 199.49±1.53 | 739.9±23.2 |
| NOTEARS | 11.79±1.05 | 99.69±1.06 | 156.8±7.7 | 22.57±1.74 | 199.65±1.52 | 741.0±23.0 |
| NOTEARS-KKTS-nzm | 6.30±0.62 | 99.04±0.96 | 174.2±7.7 | 11.75±0.96 | 198.70±1.45 | 871.8±23.5 |
| NOTEARS-KKTS | 6.21±0.64 | 99.07±0.94 | 173.5±7.7 | 11.85±0.96 | 198.76±1.46 | 874.3±23.4 |
| Abs-nzm | 27.20±1.65 | 104.05±1.24 | 34.7±2.2 | 52.60±2.09 | 209.73±1.85 | 195.2±12.2 |
| Abs | 26.67±1.60 | 103.77±1.20 | 34.9±2.0 | 52.18±1.89 | 208.66±1.62 | 202.7±12.4 |
| Abs-KKTS-nzm | 12.33±1.03 | 99.78±0.96 | 50.3±2.2 | 24.82±1.53 | 201.44±1.57 | 321.5±12.5 |
| Abs-KKTS | 12.25±1.04 | 99.46±0.95 | 50.9±2.0 | 25.45±1.40 | 201.45±1.54 | 334.0±12.7 |
| FGS-nzm | 83.28±5.61 | 196.78±8.01 | 2.6±0.2 | 114.07±8.35 | 321.52±11.14 | 5.1±0.5 |
| FGS | 83.28±5.61 | 196.78±8.01 | 2.5±0.2 | 114.07±8.35 | 321.52±11.14 | 5.1±0.5 |
| FGS-KKTS-nzm | 19.58±1.78 | 102.32±1.23 | 16.6±0.3 | 36.74±3.62 | 208.68±2.22 | 124.7±2.4 |
| FGS-KKTS | 20.31±1.73 | 102.31±1.15 | 15.7±0.3 | 36.89±3.74 | 208.55±2.18 | 122.1±2.5 |

Table 3: Effect of mean subtraction ('nzm' means nonzero mean) on ER4 graphs with Gumbel noise

| | $d = 10$ | | | $d = 30$ | | |
|---|---|---|---|---|---|---|
| | SHD | nnz | time (sec) | SHD | nnz | time (sec) |
| NOTEARS-nzm | 2.47±0.26 | 19.11±0.31 | 3.3±0.1 | 7.49±0.99 | 60.47±0.79 | 68.9±3.2 |
| NOTEARS | 2.00±0.26 | 19.24±0.32 | 2.6±0.1 | 6.11±0.89 | 60.59±0.76 | 49.9±3.6 |
| NOTEARS-KKTS-nzm | 1.37±0.16 | 19.33±0.32 | 3.4±0.1 | 3.23±0.49 | 59.97±0.73 | 73.0±3.2 |
| NOTEARS-KKTS | 0.94±0.15 | 19.42±0.30 | 2.8±0.1 | 3.07±0.54 | 60.47±0.72 | 54.1±3.6 |
| Abs-nzm | 4.25±0.42 | 19.52±0.35 | 1.2±0.1 | 15.33±1.28 | 64.34±0.96 | 13.1±0.8 |
| Abs | 3.58±0.42 | 19.55±0.35 | 1.0±0.1 | 13.27±1.07 | 63.52±0.87 | 11.2±0.7 |
| Abs-KKTS-nzm | 1.67±0.22 | 19.30±0.32 | 1.3±0.1 | 6.15±0.83 | 60.90±0.75 | 17.2±0.8 |
| Abs-KKTS | 1.14±0.18 | 19.36±0.31 | 1.1±0.1 | 5.21±0.68 | 60.87±0.76 | 15.2±0.7 |
| FGS-nzm | 12.29±0.66 | 27.85±0.83 | 0.5±0.0 | 53.42±3.56 | 119.62±4.82 | 1.3±0.1 |
| FGS | 12.29±0.66 | 27.85±0.83 | 0.5±0.0 | 53.42±3.56 | 119.62±4.82 | 1.3±0.1 |
| FGS-KKTS-nzm | 3.97±0.46 | 18.75±0.33 | 0.7±0.0 | 14.75±1.48 | 62.64±0.92 | 4.9±0.1 |
| FGS-KKTS | 3.58±0.45 | 19.12±0.31 | 0.7±0.0 | 12.36±1.38 | 62.05±0.85 | 4.8±0.1 |
| | $d = 50$ | | | $d = 100$ | | |
| | SHD | nnz | time (sec) | SHD | nnz | time (sec) |
| NOTEARS-nzm | 12.45±1.19 | 100.58±1.16 | 205.7±6.4 | 23.73±1.90 | 201.65±1.63 | 918.6±18.2 |
| NOTEARS | 12.05±1.28 | 101.03±1.20 | 169.4±7.6 | 22.97±1.92 | 202.58±1.61 | 768.4±19.5 |
| NOTEARS-KKTS-nzm | 6.57±0.76 | 100.50±1.13 | 223.8±6.5 | 11.46±1.09 | 200.99±1.45 | 1066.1±18.5 |
| NOTEARS-KKTS | 5.34±0.69 | 100.16±1.09 | 187.8±7.7 | 10.71±1.08 | 201.68±1.41 | 922.2±19.6 |
| Abs-nzm | 27.49±1.67 | 108.09±1.43 | 45.5±3.1 | 53.92±2.23 | 217.77±1.86 | 222.6±12.9 |
| Abs | 25.16±1.62 | 107.75±1.41 | 40.9±2.7 | 51.18±2.19 | 216.54±1.88 | 169.8±9.7 |
| Abs-KKTS-nzm | 9.98±0.91 | 101.67±1.15 | 63.4±3.1 | 20.28±1.47 | 204.09±1.70 | 365.7±12.8 |
| Abs-KKTS | 9.67±0.92 | 101.62±1.15 | 58.6±2.7 | 19.20±1.44 | 204.36±1.61 | 312.2±9.9 |
| FGS-nzm | 76.40±4.77 | 184.04±6.73 | 2.2±0.1 | 110.21±6.39 | 312.84±8.48 | 4.2±0.2 |
| FGS | 76.40±4.77 | 184.04±6.73 | 2.2±0.1 | 110.21±6.39 | 312.84±8.48 | 4.2±0.2 |
| FGS-KKTS-nzm | 22.77±1.82 | 104.63±1.37 | 17.0±0.4 | 34.17±2.71 | 209.42±2.06 | 135.0±3.7 |
| FGS-KKTS | 19.48±1.62 | 104.23±1.26 | 18.5±0.4 | 31.32±2.64 | 209.63±1.98 | 136.3±3.9 |

## C.4 Ablation study of KKT-informed local search

We also conduct an ablation study on KKT-informed local search by controlling which local search operations are performed. We test local search without reducing unnecessary constraints ('-noReduce'), without reversing edges ('-noReverse'), and full local search on the ER4-Gumbel case. As shown in Figure 4 (with numerical values shown in Table 4), NOTEARS-KKTS-noReverse outperforms NOTEARS-KKTS-noReduce in terms of SHD, while the opposite is true for Abs-KKTS-noReverse and Abs-KKTS-noReduce. Moreover, they are both worse than the full local search, showing that they are necessary and complement each other. In line with the discussion in Section 5, we hypothesize that Abs benefits more from reversing edges than NOTEARS because Abs by itself suffers more from poorer local minima. Time-wise, the full local search takes only slightly longer than the other methods depicted.

Figure 4: SHD and solution time of KKTS combinations without reducing unnecessary constraints ('-noReduce') and without reversing edges ('-noReverse').

Table 4: Results of KKTS combinations without reducing unnecessary constraints ('-noReduce') and without reversing edges ('-noReverse') on ER4 graphs with Gumbel noise.

|  | $d = 10$ | | | $d = 30$ | | |
|---|---|---|---|---|---|---|
|  | SHD | nnz | time (sec) | SHD | nnz | time (sec) |
| NOTEARS | 2.00±0.26 | 19.24±0.32 | 2.6±0.1 | 6.11±0.89 | 60.59±0.76 | 49.9±3.6 |
| NOTEARS-KKTS | 0.94±0.15 | 19.42±0.30 | 2.8±0.1 | 3.07±0.54 | 60.47±0.72 | 54.1±3.6 |
| NOTEARS-KKTS-noReduce | 1.79±0.23 | 19.02±0.30 | 2.6±0.1 | 4.80±0.77 | 59.26±0.69 | 49.0±3.5 |
| NOTEARS-KKTS-noReverse | 1.93±0.24 | 19.24±0.30 | 2.7±0.1 | 4.23±0.58 | 60.70±0.73 | 49.5±3.5 |
| Abs | 3.58±0.42 | 19.55±0.35 | 1.0±0.1 | 13.27±1.07 | 63.52±0.87 | 11.2±0.7 |
| Abs-KKTS | 1.14±0.18 | 19.36±0.31 | 1.1±0.1 | 5.21±0.68 | 60.87±0.76 | 15.2±0.7 |
| Abs-KKTS-noReduce | 2.54±0.35 | 18.82±0.31 | 1.0±0.1 | 9.06±0.97 | 59.59±0.75 | 11.5±0.7 |
| Abs-KKTS-noReverse | 3.39±0.38 | 19.51±0.34 | 1.1±0.1 | 11.13±0.92 | 63.74±0.83 | 11.4±0.7 |

|  | $d = 50$ | | | $d = 100$ | | |
|---|---|---|---|---|---|---|
|  | SHD | nnz | time (sec) | SHD | nnz | time (sec) |
| NOTEARS | 12.05±1.28 | 101.03±1.20 | 169.4±7.6 | 22.97±1.92 | 202.58±1.61 | 768.4±19.5 |
| NOTEARS-KKTS | 5.34±0.69 | 100.16±1.09 | 187.8±7.7 | 10.71±1.08 | 201.68±1.41 | 922.2±19.6 |
| NOTEARS-KKTS-noReduce | 9.62±1.08 | 98.05±1.03 | 165.3±7.4 | 16.99±1.52 | 196.47±1.39 | 776.2±19.9 |
| NOTEARS-KKTS-noReverse | 6.72±0.76 | 100.69±1.07 | 165.1±7.4 | 12.31±1.15 | 202.03±1.41 | 781.3±19.9 |
| Abs | 25.16±1.62 | 107.75±1.41 | 40.9±2.7 | 51.18±2.19 | 216.54±1.88 | 169.8±9.7 |
| Abs-KKTS | 9.67±0.92 | 101.62±1.15 | 58.6±2.7 | 19.20±1.44 | 204.36±1.61 | 312.2±9.9 |
| Abs-KKTS-noReduce | 15.60±1.27 | 98.73±1.12 | 41.2±2.7 | 30.99±1.88 | 198.78±1.60 | 197.6±11.2 |
| Abs-KKTS-noReverse | 19.88±1.22 | 107.66±1.25 | 41.4±2.7 | 38.84±1.65 | 216.96±1.73 | 197.9±11.1 |

## C.5  Additional results

Figures 5–7 show SHD and running time results in the same manner as Figure 1 for all tested combinations of SEM noise type (Gaussian, Gumbel, exponential), graph type (ER2, ER4, SF4), and $n = 1000$. The patterns discussed in Section 5 are quite similar across the three noise types.

Figure 5: Structural Hamming distances (SHD) and solution times for SEMs with Gaussian noise and $n = 1000$. Red lines overlap with orange in the SF4 SHD plot.

Figure 6: Structural Hamming distances (SHD) and solution times for SEMs with Gumbel noise and $n = 1000$. Red lines overlap with orange in the SF4 SHD plot.

In response to a reviewer comment, we performed a quick comparison between the original GES algorithm [7] and its FGS implementation [22]. As seen in Figure 8, not only is FGS faster than GES as expected, but its SHD is also much better. After applying KKTS however, FGS-KKTS and GES-KKTS are similar.

Tables 5–13 show the same results as Figures 5–7 in tabular form. In addition, results for CAM [6] are also shown and are seen to be less competitive in the linear SEM setting tested here. Nevertheless, like the other -KKTS combinations, CAM-KKTS succeeds in improving the SHDs of CAM, by large factors in some cases.

Figure 7: Structural Hamming distances (SHD) and solution times for SEMs with exponential noise and $n = 1000$. Red lines overlap with orange in the SF4 SHD plot.

Figure 8: SHD and solution time for the same $n = 2d$ setting as in Figure 2, showing that FGS is both faster and more accurate than GES.

Tables 14–16 show the same results as Figure 2 in tabular form.

Table 5: Results (mean ± standard error over 100 trials) on ER2 graphs with Gaussian noise, $n = 1000$.

| | $d = 10$ | | | $d = 30$ | | |
|---|---|---|---|---|---|---|
| | SHD | nnz | time (sec) | SHD | nnz | time (sec) |
| NOTEARS | 0.78±0.15 | 10.04±0.31 | 1.1±0.1 | 0.90±0.14 | 29.41±0.52 | 8.6±0.8 |
| NOTEARS-KKTS | 0.54±0.13 | 10.13±0.32 | 1.2±0.1 | 0.58±0.10 | 29.50±0.53 | 10.5±0.8 |
| NOTEARS-1e-5 | 0.83±0.15 | 10.00±0.31 | 0.7±0.0 | 1.19±0.16 | 29.18±0.52 | 3.9±0.2 |
| NOTEARS-1e-5-KKTS | 0.55±0.13 | 10.12±0.32 | 0.9±0.0 | 0.66±0.11 | 29.47±0.53 | 5.9±0.2 |
| Abs | 0.91±0.17 | 10.12±0.32 | 0.4±0.0 | 2.75±0.34 | 29.54±0.55 | 2.3±0.2 |
| Abs-KKTS | 0.39±0.09 | 10.12±0.31 | 0.5±0.0 | 1.00±0.16 | 29.37±0.53 | 4.2±0.2 |
| FGS | 3.36±0.34 | 12.04±0.59 | 0.5±0.0 | 7.65±0.56 | 32.87±0.88 | 0.7±0.0 |
| FGS-KKTS | 0.88±0.21 | 10.15±0.31 | 0.6±0.0 | 1.15±0.24 | 29.49±0.53 | 2.6±0.1 |
| MMHC | 5.20±0.34 | 9.38±0.23 | 0.4±0.0 | 11.71±0.46 | 28.13±0.42 | 0.8±0.0 |
| MMHC-KKTS | 1.25±0.21 | 9.92±0.31 | 0.5±0.0 | 3.34±0.42 | 29.56±0.57 | 2.0±0.0 |
| PC | 5.94±0.36 | 12.21±0.27 | 2.8±0.1 | 14.42±0.53 | 36.83±0.47 | 3.2±0.0 |
| PC-KKTS | 1.05±0.20 | 10.05±0.31 | 3.0±0.1 | 2.37±0.36 | 29.46±0.54 | 5.0±0.1 |
| Search | 2.18±0.31 | 9.89±0.29 | 0.2±0.0 | 7.45±0.58 | 28.76±0.52 | 2.7±0.1 |
| CAM | 7.79±0.54 | 12.75±0.49 | 13.8±0.4 | 20.83±0.90 | 37.78±0.85 | 68.6±1.2 |
| CAM-KKTS | 1.41±0.22 | 10.19±0.33 | 14.0±0.4 | 3.81±0.49 | 29.49±0.54 | 71.0±1.2 |

| | $d = 50$ | | | $d = 100$ | | |
|---|---|---|---|---|---|---|
| | SHD | nnz | time (sec) | SHD | nnz | time (sec) |
| NOTEARS | 1.83±0.28 | 50.07±0.68 | 29.0±2.5 | 3.18±0.40 | 97.21±0.91 | 175.2±10.9 |
| NOTEARS-KKTS | 1.05±0.16 | 50.18±0.68 | 36.8±2.5 | 1.59±0.22 | 97.51±0.89 | 232.6±11.1 |
| NOTEARS-1e-5 | 2.31±0.26 | 49.79±0.69 | 11.8±0.7 | 3.85±0.35 | 96.64±0.90 | 62.3±3.1 |
| NOTEARS-1e-5-KKTS | 1.23±0.16 | 50.21±0.69 | 19.4±0.8 | 1.79±0.21 | 97.48±0.90 | 117.6±3.4 |
| Abs | 6.25±0.53 | 50.90±0.77 | 6.2±0.4 | 13.31±0.71 | 98.53±1.07 | 35.0±2.2 |
| Abs-KKTS | 1.87±0.25 | 50.03±0.70 | 13.5±0.5 | 3.25±0.30 | 97.12±0.93 | 91.1±2.5 |
| FGS | 10.65±0.88 | 54.85±1.32 | 0.8±0.0 | 20.33±0.84 | 104.42±1.38 | 1.3±0.0 |
| FGS-KKTS | 1.64±0.23 | 50.34±0.70 | 8.7±0.1 | 2.52±0.28 | 97.88±0.91 | 59.2±0.9 |
| MMHC | 18.66±0.60 | 46.42±0.52 | 1.0±0.0 | 36.71±0.81 | 93.25±0.77 | 2.3±0.0 |
| MMHC-KKTS | 6.26±0.59 | 50.18±0.74 | 5.6±0.1 | 10.71±0.91 | 98.14±0.99 | 32.7±0.5 |
| PC | 21.99±0.65 | 61.79±0.64 | 3.8±0.1 | 40.27±0.86 | 123.45±0.92 | 5.1±0.1 |
| PC-KKTS | 3.71±0.37 | 50.38±0.70 | 9.9±0.1 | 6.76±0.58 | 97.74±0.94 | 49.7±0.8 |
| Search | 13.81±0.86 | 48.70±0.68 | 11.2±0.3 | 27.89±1.08 | 95.02±0.94 | 120.2±3.0 |
| CAM | 34.90±0.98 | 64.16±1.03 | 127.9±2.1 | 65.40±1.31 | 123.84±1.29 | 253.8±2.7 |
| CAM-KKTS | 7.24±0.66 | 50.71±0.75 | 142.5±2.0 | 10.80±0.84 | 98.61±0.97 | 360.8±4.1 |

Table 6: Results (mean $\pm$ standard error over 100 trials) on ER4 graphs with Gaussian noise, $n = 1000$.

| | $d = 10$ | | | $d = 30$ | | |
|---|---|---|---|---|---|---|
| | SHD | nnz | time (sec) | SHD | nnz | time (sec) |
| NOTEARS | 3.61±0.36 | 18.08±0.29 | 2.8±0.2 | 7.42±0.81 | 57.97±0.74 | 45.9±3.4 |
| NOTEARS-KKTS | 1.87±0.20 | 18.37±0.30 | 2.9±0.2 | 4.70±0.58 | 58.18±0.72 | 49.5±3.4 |
| NOTEARS-1e-5 | 3.85±0.37 | 17.89±0.30 | 1.9±0.1 | 9.02±0.87 | 57.52±0.76 | 16.4±1.0 |
| NOTEARS-1e-5-KKTS | 1.95±0.22 | 18.35±0.30 | 2.1±0.1 | 5.00±0.57 | 58.08±0.70 | 20.1±1.0 |
| Abs | 4.52±0.41 | 18.16±0.30 | 0.9±0.1 | 15.06±1.06 | 60.39±0.81 | 9.8±0.6 |
| Abs-KKTS | 2.54±0.27 | 18.19±0.31 | 1.1±0.1 | 6.14±0.56 | 58.23±0.72 | 13.4±0.6 |
| FGS | 13.48±0.74 | 28.44±0.81 | 0.5±0.0 | 53.21±3.30 | 118.38±4.48 | 1.3±0.1 |
| FGS-KKTS | 4.92±0.47 | 17.79±0.30 | 0.7±0.0 | 15.03±1.38 | 59.72±0.87 | 4.5±0.1 |
| MMHC | 15.51±0.50 | 11.90±0.17 | 0.5±0.0 | 41.15±1.18 | 35.85±0.43 | 0.9±0.0 |
| MMHC-KKTS | 6.53±0.53 | 17.50±0.32 | 0.6±0.0 | 22.65±1.57 | 59.48±0.92 | 3.0±0.0 |
| PC | 16.35±0.50 | 15.26±0.23 | 2.4±0.0 | 46.28±1.23 | 45.46±0.46 | 3.2±0.0 |
| PC-KKTS | 6.17±0.52 | 17.26±0.32 | 2.5±0.0 | 20.00±1.44 | 59.15±0.85 | 5.6±0.1 |
| Search | 9.12±0.59 | 15.75±0.31 | 0.3±0.0 | 34.61±1.37 | 51.10±0.70 | 6.5±0.1 |
| CAM | 19.06±0.64 | 22.84±0.37 | 12.9±0.3 | 56.80±1.70 | 77.53±1.06 | 65.7±1.1 |
| CAM-KKTS | 6.64±0.57 | 17.42±0.30 | 13.1±0.3 | 19.55±1.45 | 59.48±0.84 | 70.8±1.2 |

| | $d = 50$ | | | $d = 100$ | | |
|---|---|---|---|---|---|---|
| | SHD | nnz | time (sec) | SHD | nnz | time (sec) |
| NOTEARS | 11.79±1.05 | 99.69±1.06 | 156.8±7.7 | 22.57±1.74 | 199.65±1.52 | 741.0±23.0 |
| NOTEARS-KKTS | 6.21±0.64 | 99.07±0.94 | 173.5±7.7 | 11.85±0.96 | 198.76±1.46 | 874.3±23.4 |
| NOTEARS-1e-5 | 13.25±0.99 | 98.51±1.06 | 52.2±2.2 | 23.16±1.57 | 197.89±1.48 | 264.9±10.1 |
| NOTEARS-1e-5-KKTS | 6.36±0.55 | 98.66±0.95 | 68.1±2.2 | 12.13±0.92 | 198.56±1.46 | 391.8±10.5 |
| Abs | 26.67±1.60 | 103.77±1.20 | 34.9±2.0 | 52.18±1.89 | 208.66±1.62 | 202.7±12.4 |
| Abs-KKTS | 12.25±1.04 | 99.46±0.95 | 50.9±2.0 | 25.45±1.40 | 201.45±1.54 | 334.0±12.7 |
| FGS | 83.28±5.61 | 196.78±8.01 | 2.5±0.2 | 114.07±8.35 | 321.52±11.14 | 5.1±0.5 |
| FGS-KKTS | 20.31±1.73 | 102.31±1.15 | 15.7±0.3 | 36.89±3.74 | 208.55±2.18 | 122.1±2.5 |
| MMHC | 66.13±1.50 | 61.29±0.55 | 2.3±0.1 | 120.66±2.06 | 128.66±1.10 | 13.6±0.3 |
| MMHC-KKTS | 36.27±2.05 | 104.06±1.32 | 11.1±0.2 | 69.24±3.35 | 213.52±2.14 | 87.0±1.2 |
| PC | 74.99±1.62 | 77.16±0.56 | 4.2±0.1 | 132.77±2.52 | 152.49±0.93 | 8.5±0.2 |
| PC-KKTS | 36.73±2.14 | 103.28±1.23 | 14.0±0.2 | 65.88±3.99 | 212.44±2.27 | 75.2±1.1 |
| Search | 56.23±1.66 | 89.33±0.98 | 34.3±0.6 | 106.24±2.58 | 181.97±1.41 | 484.0±7.1 |
| CAM | 91.13±2.02 | 129.64±1.43 | 130.2±1.9 | 159.91±3.11 | 247.67±1.86 | 271.4±3.2 |
| CAM-KKTS | 34.76±1.98 | 104.17±1.23 | 146.1±2.1 | 68.13±3.71 | 212.93±2.19 | 388.3±5.7 |

Table 7: Results (mean $\pm$ standard error over 100 trials) on SF4 graphs with Gaussian noise, $n = 1000$.

| | $d = 10$ | | | $d = 30$ | | |
|---|---|---|---|---|---|---|
| | SHD | nnz | time (sec) | SHD | nnz | time (sec) |
| NOTEARS | 0.48±0.12 | 13.67±0.18 | 0.7±0.0 | 0.99±0.16 | 49.91±0.33 | 4.5±0.2 |
| NOTEARS-KKTS | 0.29±0.09 | 13.64±0.16 | 0.8±0.0 | 0.76±0.10 | 49.95±0.33 | 7.0±0.2 |
| NOTEARS-1e-5 | 0.48±0.12 | 13.67±0.18 | 0.5±0.0 | 1.00±0.16 | 49.90±0.33 | 3.2±0.1 |
| NOTEARS-1e-5-KKTS | 0.29±0.09 | 13.64±0.16 | 0.6±0.0 | 0.77±0.10 | 49.94±0.33 | 5.6±0.1 |
| Abs | 0.54±0.15 | 13.62±0.16 | 0.2±0.0 | 2.29±0.39 | 50.03±0.35 | 1.9±0.1 |
| Abs-KKTS | 0.35±0.12 | 13.63±0.16 | 0.4±0.0 | 1.01±0.12 | 49.93±0.33 | 4.3±0.1 |
| FGS | 4.42±0.55 | 17.25±0.71 | 0.4±0.0 | 22.37±1.96 | 65.48±2.39 | 0.7±0.0 |
| FGS-KKTS | 0.78±0.17 | 13.58±0.16 | 0.6±0.0 | 2.43±0.49 | 50.05±0.36 | 3.2±0.0 |
| MMHC | 6.53±0.37 | 11.59±0.14 | 0.5±0.0 | 27.73±0.68 | 34.48±0.41 | 11.8±3.9 |
| MMHC-KKTS | 0.93±0.20 | 13.58±0.17 | 0.6±0.0 | 4.36±0.66 | 49.78±0.37 | 13.4±3.9 |
| PC | 6.44±0.33 | 12.35±0.17 | 2.3±0.0 | 29.09±0.58 | 36.11±0.39 | 9.2±2.5 |
| PC-KKTS | 0.91±0.20 | 13.54±0.16 | 2.4±0.0 | 4.05±0.65 | 49.83±0.36 | 10.9±2.5 |
| Search | 1.78±0.31 | 13.28±0.17 | 0.3±0.0 | 11.11±0.98 | 49.18±0.43 | 3.6±0.1 |
| CAM | 13.70±0.76 | 19.90±0.53 | 11.5±0.2 | 46.72±1.35 | 57.86±1.03 | 51.5±0.7 |
| CAM-KKTS | 1.36±0.27 | 13.46±0.18 | 11.6±0.2 | 4.09±0.63 | 49.92±0.36 | 53.1±0.7 |
| | $d = 50$ | | | $d = 100$ | | |
| | SHD | nnz | time (sec) | SHD | nnz | time (sec) |
| NOTEARS | 1.44±0.39 | 87.81±0.37 | 17.3±0.9 | 3.28±0.88 | 183.95±0.60 | 110.3±5.6 |
| NOTEARS-KKTS | 0.74±0.12 | 87.74±0.37 | 27.7±0.9 | 2.38±0.57 | 183.80±0.55 | 195.0±5.9 |
| NOTEARS-1e-5 | 1.50±0.39 | 87.80±0.37 | 10.6±0.4 | 3.43±0.89 | 183.93±0.60 | 60.8±3.6 |
| NOTEARS-1e-5-KKTS | 0.75±0.12 | 87.74±0.37 | 21.2±0.4 | 2.42±0.58 | 183.78±0.55 | 146.2±3.9 |
| Abs | 5.14±0.73 | 88.45±0.47 | 7.4±0.4 | 15.40±1.74 | 186.68±0.74 | 50.7±4.1 |
| Abs-KKTS | 1.62±0.25 | 87.67±0.39 | 17.4±0.5 | 4.42±0.79 | 183.09±0.54 | 133.7±4.3 |
| FGS | 42.94±2.79 | 107.87±3.31 | 1.1±0.0 | 89.18±4.25 | 193.06±4.79 | 2.3±0.1 |
| FGS-KKTS | 5.61±0.95 | 87.94±0.50 | 11.5±0.1 | 11.42±1.51 | 181.64±0.79 | 92.7±1.1 |
| MMHC | 54.89±1.00 | 54.53±0.66 | 21.0±4.8 | 121.35±1.44 | 114.06±1.02 | 194.7±106.3 |
| MMHC-KKTS | 10.07±1.13 | 87.06±0.53 | 28.6±4.8 | 20.14±2.03 | 181.63±0.90 | 255.1±106.4 |
| PC | 54.98±0.80 | 58.14±0.68 | 10.1±1.4 | 118.74±1.12 | 120.68±0.98 | 54.4±23.1 |
| PC-KKTS | 8.34±0.93 | 86.42±0.56 | 16.7±1.4 | 21.56±2.08 | 180.15±0.93 | 100.7±23.1 |
| Search | 23.74±1.84 | 85.59±0.72 | 14.1±0.2 | 50.30±2.96 | 178.68±0.96 | 152.7±2.7 |
| CAM | 82.51±2.01 | 89.62±1.43 | 91.0±1.0 | 157.53±2.44 | 160.91±1.95 | 223.3±1.9 |
| CAM-KKTS | 11.91±1.36 | 88.47±0.65 | 98.8±1.0 | 24.74±2.16 | 184.40±0.97 | 288.3±2.4 |

Table 8: Results (mean $\pm$ standard error over 100 trials) on ER2 graphs with Gumbel noise, $n = 1000$.

| | $d = 10$ | | | $d = 30$ | | |
|---|---|---|---|---|---|---|
| | SHD | nnz | time (sec) | SHD | nnz | time (sec) |
| NOTEARS | 0.40±0.10 | 9.49±0.27 | 1.4±0.2 | 0.43±0.10 | 29.71±0.56 | 8.2±0.5 |
| NOTEARS-KKTS | 0.07±0.04 | 9.56±0.27 | 1.5±0.2 | 0.12±0.03 | 29.69±0.56 | 10.5±0.5 |
| NOTEARS-1e-5 | 0.45±0.10 | 9.45±0.27 | 0.8±0.1 | 0.63±0.12 | 29.52±0.55 | 4.0±0.2 |
| NOTEARS-1e-5-KKTS | 0.15±0.06 | 9.55±0.27 | 0.9±0.1 | 0.16±0.04 | 29.69±0.56 | 6.2±0.2 |
| Abs | 0.59±0.13 | 9.58±0.26 | 0.3±0.0 | 2.59±0.28 | 30.72±0.60 | 2.4±0.2 |
| Abs-KKTS | 0.18±0.08 | 9.52±0.27 | 0.5±0.0 | 0.23±0.06 | 29.68±0.56 | 4.7±0.2 |
| FGS | 2.90±0.27 | 10.63±0.45 | 0.4±0.0 | 7.18±0.64 | 33.09±0.98 | 0.6±0.0 |
| FGS-KKTS | 0.41±0.10 | 9.58±0.28 | 0.6±0.0 | 0.80±0.18 | 29.72±0.57 | 2.8±0.1 |
| MMHC | 4.33±0.29 | 8.82±0.20 | 0.4±0.0 | 11.36±0.51 | 28.05±0.41 | 0.8±0.0 |
| MMHC-KKTS | 1.13±0.21 | 9.54±0.27 | 0.5±0.0 | 2.72±0.38 | 30.27±0.61 | 2.2±0.0 |
| PC | 5.29±0.30 | 12.18±0.26 | 2.2±0.0 | 12.75±0.49 | 35.72±0.41 | 2.6±0.0 |
| PC-KKTS | 0.91±0.20 | 9.51±0.27 | 2.3±0.0 | 1.72±0.29 | 30.09±0.61 | 4.2±0.1 |
| Search | 1.92±0.30 | 9.50±0.27 | 0.2±0.0 | 6.25±0.48 | 29.71±0.57 | 3.3±0.1 |
| CAM | 9.84±0.43 | 12.93±0.42 | 10.4±0.2 | 30.88±0.97 | 41.95±1.01 | 48.7±0.7 |
| CAM-KKTS | 1.23±0.21 | 9.70±0.29 | 10.5±0.2 | 4.66±0.58 | 30.57±0.65 | 50.1±0.7 |
| | $d = 50$ | | | $d = 100$ | | |
| | SHD | nnz | time (sec) | SHD | nnz | time (sec) |
| NOTEARS | 1.90±0.31 | 50.89±0.71 | 39.7±3.6 | 2.91±0.40 | 100.61±0.98 | 220.2±12.5 |
| NOTEARS-KKTS | 0.62±0.14 | 51.09±0.70 | 49.3±3.7 | 1.02±0.19 | 100.68±1.00 | 301.6±12.8 |
| NOTEARS-1e-5 | 2.41±0.33 | 50.72±0.73 | 14.1±0.8 | 3.63±0.43 | 100.32±0.98 | 89.0±4.7 |
| NOTEARS-1e-5-KKTS | 0.86±0.18 | 51.11±0.71 | 23.9±0.9 | 1.22±0.21 | 100.69±0.99 | 171.6±5.1 |
| Abs | 6.67±0.57 | 52.60±0.78 | 7.7±0.6 | 13.27±0.98 | 103.49±1.30 | 40.2±2.5 |
| Abs-KKTS | 1.31±0.30 | 51.23±0.72 | 17.1±0.7 | 2.55±0.33 | 101.25±1.02 | 118.1±3.0 |
| FGS | 10.83±0.67 | 54.93±1.14 | 0.8±0.0 | 20.47±0.76 | 106.67±1.37 | 1.2±0.0 |
| FGS-KKTS | 1.83±0.35 | 51.27±0.74 | 10.3±0.2 | 2.33±0.37 | 101.12±1.03 | 84.0±1.1 |
| MMHC | 19.93±0.59 | 48.54±0.52 | 1.0±0.0 | 40.88±0.86 | 100.67±0.72 | 2.5±0.0 |
| MMHC-KKTS | 6.06±0.60 | 51.92±0.76 | 6.8±0.1 | 10.22±0.77 | 102.97±1.09 | 47.2±0.6 |
| PC | 23.07±0.68 | 62.33±0.60 | 3.3±0.1 | 44.31±0.88 | 125.32±0.87 | 4.6±0.1 |
| PC-KKTS | 4.13±0.49 | 51.58±0.78 | 9.1±0.2 | 7.22±0.85 | 102.45±1.18 | 48.4±0.7 |
| Search | 14.61±0.81 | 51.13±0.74 | 16.4±0.4 | 29.84±1.26 | 100.76±1.05 | 269.1±4.2 |
| CAM | 51.11±1.13 | 71.25±1.19 | 90.3±0.8 | 100.67±1.61 | 140.15±1.55 | 210.9±1.4 |
| CAM-KKTS | 8.51±0.77 | 52.80±0.82 | 96.0±0.8 | 22.19±1.62 | 106.30±1.25 | 258.2±1.5 |

Table 9: Results (mean $\pm$ standard error over 100 trials) on ER4 graphs with Gumbel noise, $n = 1000$.

| | $d = 10$ | | | $d = 30$ | | |
|---|---|---|---|---|---|---|
| | SHD | nnz | time (sec) | SHD | nnz | time (sec) |
| NOTEARS | 2.00±0.26 | 19.24±0.32 | 2.6±0.1 | 6.11±0.89 | 60.59±0.76 | 49.9±3.6 |
| NOTEARS-KKTS | 0.94±0.15 | 19.42±0.30 | 2.8±0.1 | 3.07±0.54 | 60.47±0.72 | 54.1±3.6 |
| NOTEARS-1e-5 | 2.10±0.26 | 19.15±0.32 | 1.8±0.1 | 7.21±0.88 | 59.97±0.76 | 18.9±1.2 |
| NOTEARS-1e-5-KKTS | 0.94±0.15 | 19.42±0.30 | 1.9±0.1 | 3.38±0.54 | 60.42±0.72 | 22.9±1.1 |
| Abs | 3.58±0.42 | 19.55±0.35 | 1.0±0.1 | 13.27±1.07 | 63.52±0.87 | 11.2±0.7 |
| Abs-KKTS | 1.14±0.18 | 19.36±0.31 | 1.1±0.1 | 5.21±0.68 | 60.87±0.76 | 15.2±0.7 |
| FGS | 12.29±0.66 | 27.85±0.83 | 0.5±0.0 | 53.42±3.56 | 119.62±4.82 | 1.3±0.1 |
| FGS-KKTS | 3.58±0.45 | 19.12±0.31 | 0.7±0.0 | 12.36±1.38 | 62.05±0.85 | 4.8±0.1 |
| MMHC | 14.88±0.52 | 12.33±0.17 | 0.4±0.0 | 42.18±1.04 | 35.87±0.38 | 2.1±0.0 |
| MMHC-KKTS | 5.11±0.49 | 18.85±0.29 | 0.6±0.0 | 23.45±1.55 | 64.07±0.90 | 4.5±0.1 |
| PC | 16.01±0.52 | 15.25±0.24 | 2.8±0.1 | 47.03±1.16 | 45.27±0.47 | 3.7±0.0 |
| PC-KKTS | 5.41±0.50 | 18.62±0.29 | 3.0±0.1 | 21.45±1.56 | 64.64±0.98 | 7.0±0.1 |
| Search | 8.31±0.58 | 16.44±0.33 | 0.4±0.0 | 31.97±1.15 | 54.64±0.69 | 7.3±0.1 |
| CAM | 20.66±0.47 | 23.68±0.36 | 9.9±0.1 | 65.42±1.36 | 81.94±0.96 | 46.3±0.5 |
| CAM-KKTS | 5.62±0.49 | 18.66±0.31 | 10.0±0.1 | 22.34±1.42 | 63.18±0.92 | 48.5±0.5 |

| | $d = 50$ | | | $d = 100$ | | |
|---|---|---|---|---|---|---|
| | SHD | nnz | time (sec) | SHD | nnz | time (sec) |
| NOTEARS | 12.05±1.28 | 101.03±1.20 | 169.4±7.6 | 22.97±1.92 | 202.58±1.61 | 768.4±19.5 |
| NOTEARS-KKTS | 5.34±0.69 | 100.16±1.09 | 187.8±7.7 | 10.71±1.08 | 201.68±1.41 | 922.2±19.6 |
| NOTEARS-1e-5 | 12.99±1.19 | 100.38±1.18 | 61.8±2.9 | 23.66±1.81 | 201.62±1.60 | 313.9±11.2 |
| NOTEARS-1e-5-KKTS | 6.14±0.72 | 100.39±1.10 | 80.1±2.9 | 11.19±1.06 | 201.45±1.37 | 466.3±11.3 |
| Abs | 25.16±1.62 | 107.75±1.41 | 40.9±2.7 | 51.18±2.19 | 216.54±1.88 | 169.8±9.7 |
| Abs-KKTS | 9.67±0.92 | 101.62±1.15 | 58.6±2.7 | 19.20±1.44 | 204.36±1.61 | 312.2±9.9 |
| FGS | 76.40±4.77 | 184.04±6.73 | 2.2±0.1 | 110.21±6.39 | 312.84±8.48 | 4.2±0.2 |
| FGS-KKTS | 19.48±1.62 | 104.23±1.26 | 18.5±0.4 | 31.32±2.64 | 209.63±1.98 | 136.3±3.9 |
| MMHC | 64.74±1.59 | 60.97±0.68 | 1.5±0.0 | 119.00±2.10 | 129.00±0.96 | 14.3±0.2 |
| MMHC-KKTS | 38.28±2.48 | 108.72±1.65 | 11.6±0.2 | 76.75±3.87 | 224.75±2.57 | 103.7±1.2 |
| PC | 74.85±1.64 | 75.99±0.61 | 5.1±0.1 | 131.01±2.41 | 153.06±0.89 | 10.4±0.1 |
| PC-KKTS | 40.53±2.58 | 108.84±1.56 | 18.7±0.3 | 67.96±3.91 | 221.17±2.49 | 115.1±1.7 |
| Search | 51.67±1.96 | 92.10±0.98 | 39.4±0.7 | 97.72±2.46 | 189.95±1.61 | 630.7±7.9 |
| CAM | 105.83±2.02 | 134.77±1.41 | 91.7±1.0 | 187.69±2.68 | 256.63±1.77 | 216.3±1.9 |
| CAM-KKTS | 47.10±2.35 | 109.91±1.54 | 100.6±1.0 | 86.06±3.72 | 224.16±2.58 | 306.3±2.9 |

Table 10: Results (mean ± standard error over 100 trials) on SF4 graphs with Gumbel noise, $n = 1000$.

| | $d = 10$ | | | $d = 30$ | | |
|---|---|---|---|---|---|---|
| | SHD | nnz | time (sec) | SHD | nnz | time (sec) |
| NOTEARS | 0.19±0.07 | 13.68±0.16 | 0.7±0.0 | 0.69±0.25 | 50.12±0.31 | 6.2±0.3 |
| NOTEARS-KKTS | 0.08±0.03 | 13.72±0.16 | 0.9±0.0 | 0.26±0.06 | 50.08±0.30 | 9.0±0.3 |
| NOTEARS-1e-5 | 0.19±0.07 | 13.68±0.16 | 0.5±0.0 | 0.69±0.25 | 50.11±0.31 | 3.9±0.2 |
| NOTEARS-1e-5-KKTS | 0.08±0.03 | 13.72±0.16 | 0.7±0.0 | 0.35±0.10 | 50.12±0.31 | 6.7±0.2 |
| Abs | 0.23±0.06 | 13.74±0.16 | 0.3±0.0 | 2.92±0.60 | 50.96±0.46 | 2.6±0.2 |
| Abs-KKTS | 0.12±0.04 | 13.72±0.16 | 0.4±0.0 | 1.07±0.33 | 50.34±0.36 | 5.4±0.2 |
| FGS | 3.76±0.52 | 16.17±0.59 | 0.4±0.0 | 26.25±2.10 | 67.17±2.42 | 0.8±0.0 |
| FGS-KKTS | 0.26±0.09 | 13.70±0.15 | 0.6±0.0 | 3.31±0.67 | 50.51±0.39 | 3.6±0.0 |
| MMHC | 6.28±0.39 | 11.66±0.14 | 0.5±0.0 | 29.08±0.80 | 33.04±0.47 | 6.5±2.5 |
| MMHC-KKTS | 1.03±0.23 | 13.90±0.18 | 0.6±0.0 | 5.32±0.86 | 51.29±0.48 | 8.4±2.5 |
| PC | 6.55±0.28 | 12.56±0.16 | 2.5±0.1 | 29.18±0.56 | 35.36±0.39 | 7.8±1.2 |
| PC-KKTS | 0.84±0.21 | 14.05±0.16 | 2.6±0.1 | 3.79±0.63 | 50.45±0.42 | 10.0±1.2 |
| Search | 1.92±0.30 | 13.53±0.17 | 0.3±0.0 | 13.86±1.37 | 50.33±0.47 | 4.4±0.1 |
| CAM | 19.95±0.63 | 22.73±0.42 | 11.3±0.2 | 64.41±1.28 | 65.37±1.02 | 50.8±0.7 |
| CAM-KKTS | 1.32±0.32 | 13.81±0.18 | 11.4±0.2 | 7.74±1.04 | 51.65±0.59 | 52.7±0.7 |

| | $d = 50$ | | | $d = 100$ | | |
|---|---|---|---|---|---|---|
| | SHD | nnz | time (sec) | SHD | nnz | time (sec) |
| NOTEARS | 1.22±0.45 | 88.35±0.40 | 24.4±1.3 | 2.53±0.57 | 184.08±0.65 | 157.4±7.4 |
| NOTEARS-KKTS | 0.43±0.12 | 88.34±0.38 | 36.9±1.4 | 1.53±0.50 | 183.66±0.59 | 262.6±7.8 |
| NOTEARS-1e-5 | 1.26±0.46 | 88.35±0.40 | 13.3±0.7 | 2.64±0.59 | 184.04±0.65 | 84.6±5.2 |
| NOTEARS-1e-5-KKTS | 0.45±0.13 | 88.35±0.39 | 25.8±0.7 | 1.41±0.49 | 183.60±0.59 | 191.6±5.7 |
| Abs | 6.29±0.84 | 90.68±0.61 | 10.2±0.7 | 11.69±1.22 | 187.94±0.94 | 72.3±5.8 |
| Abs-KKTS | 1.82±0.52 | 88.73±0.45 | 22.7±0.8 | 4.33±0.82 | 184.23±0.64 | 179.3±6.4 |
| FGS | 43.64±2.76 | 107.99±3.43 | 1.1±0.0 | 90.96±3.91 | 188.82±4.27 | 2.3±0.1 |
| FGS-KKTS | 6.79±1.36 | 89.74±0.72 | 13.5±0.2 | 19.44±2.90 | 187.95±1.55 | 109.9±1.7 |
| MMHC | 55.35±0.95 | 55.43±0.63 | 21.3±7.8 | 124.12±1.43 | 112.37±0.96 | 118.1±49.3 |
| MMHC-KKTS | 9.09±1.25 | 89.84±0.65 | 30.7±7.8 | 21.86±2.21 | 186.37±1.13 | 194.8±49.4 |
| PC | 55.07±0.98 | 58.65±0.67 | 47.6±21.6 | 123.02±1.30 | 118.71±0.93 | 36.7±18.3 |
| PC-KKTS | 8.96±1.28 | 88.93±0.66 | 56.4±21.6 | 29.16±2.99 | 186.16±1.18 | 102.1±18.4 |
| Search | 26.20±1.82 | 88.97±0.73 | 20.2±0.4 | 54.18±3.33 | 183.98±1.10 | 274.7±3.5 |
| CAM | 104.84±1.34 | 99.24±1.14 | 89.6±1.0 | 212.09±2.31 | 182.76±2.08 | 211.1±1.4 |
| CAM-KKTS | 16.81±1.92 | 92.20±0.87 | 98.4±1.0 | 35.73±3.32 | 190.62±1.37 | 284.3±2.0 |

Table 11: Results (mean $\pm$ standard error over 100 trials) on ER2 graphs with exponential noise, $n = 1000$.

| | $d = 10$ | | | $d = 30$ | | |
|---|---|---|---|---|---|---|
| | SHD | nnz | time (sec) | SHD | nnz | time (sec) |
| NOTEARS | 0.58±0.12 | 9.87±0.30 | 1.8±0.1 | 1.10±0.12 | 28.84±0.49 | 8.0±0.6 |
| NOTEARS-KKTS | 0.26±0.07 | 9.89±0.29 | 2.0±0.1 | 0.65±0.11 | 28.94±0.49 | 9.8±0.7 |
| NOTEARS-1e-5 | 0.64±0.12 | 9.80±0.29 | 1.2±0.1 | 1.43±0.14 | 28.51±0.49 | 3.9±0.2 |
| NOTEARS-1e-5-KKTS | 0.26±0.07 | 9.89±0.29 | 1.4±0.1 | 0.78±0.12 | 28.87±0.49 | 5.8±0.2 |
| Abs | 0.95±0.18 | 9.86±0.29 | 0.6±0.1 | 2.37±0.26 | 28.97±0.50 | 2.8±0.2 |
| Abs-KKTS | 0.38±0.10 | 9.83±0.29 | 0.8±0.1 | 1.05±0.16 | 28.88±0.50 | 5.4±0.3 |
| FGS | 3.81±0.41 | 11.61±0.59 | 0.4±0.0 | 6.87±0.45 | 31.59±0.75 | 0.6±0.0 |
| FGS-KKTS | 0.58±0.12 | 9.81±0.29 | 0.6±0.0 | 1.10±0.20 | 29.00±0.50 | 2.5±0.0 |
| MMHC | 4.95±0.30 | 9.06±0.21 | 0.4±0.0 | 11.46±0.46 | 27.73±0.41 | 0.6±0.0 |
| MMHC-KKTS | 1.03±0.18 | 9.82±0.30 | 0.5±0.0 | 2.90±0.35 | 29.00±0.50 | 1.8±0.0 |
| PC | 5.86±0.31 | 12.34±0.29 | 2.9±0.1 | 13.61±0.43 | 35.86±0.46 | 2.5±0.0 |
| PC-KKTS | 0.87±0.17 | 9.69±0.29 | 3.0±0.1 | 1.98±0.25 | 28.92±0.49 | 3.8±0.0 |
| Search | 2.02±0.27 | 9.48±0.28 | 0.3±0.0 | 7.29±0.55 | 28.18±0.50 | 3.1±0.2 |
| CAM | 11.48±0.43 | 14.25±0.46 | 9.5±0.1 | 32.43±0.79 | 42.58±0.78 | 45.4±0.5 |
| CAM-KKTS | 1.69±0.25 | 9.76±0.29 | 9.6±0.1 | 5.50±0.53 | 29.16±0.52 | 46.5±0.5 |

| | $d = 50$ | | | $d = 100$ | | |
|---|---|---|---|---|---|---|
| | SHD | nnz | time (sec) | SHD | nnz | time (sec) |
| NOTEARS | 1.68±0.27 | 49.26±0.66 | 29.6±2.7 | 2.92±0.28 | 98.12±0.82 | 158.4±10.1 |
| NOTEARS-KKTS | 0.98±0.17 | 49.45±0.66 | 36.8±2.7 | 1.66±0.20 | 98.39±0.82 | 215.7±10.2 |
| NOTEARS-1e-5 | 2.04±0.25 | 49.01±0.65 | 13.0±0.9 | 3.73±0.32 | 97.56±0.81 | 61.3±2.8 |
| NOTEARS-1e-5-KKTS | 1.03±0.18 | 49.40±0.65 | 20.5±0.9 | 1.88±0.22 | 98.29±0.82 | 117.8±3.0 |
| Abs | 6.11±0.47 | 50.24±0.74 | 6.9±0.5 | 14.47±0.66 | 100.20±1.02 | 30.9±1.5 |
| Abs-KKTS | 1.73±0.21 | 49.15±0.69 | 14.1±0.5 | 3.64±0.34 | 98.02±0.84 | 87.1±1.8 |
| FGS | 12.38±0.87 | 55.58±1.42 | 0.8±0.0 | 19.81±0.75 | 105.59±1.17 | 1.2±0.0 |
| FGS-KKTS | 1.92±0.29 | 49.48±0.65 | 7.9±0.1 | 2.77±0.37 | 98.76±0.83 | 58.1±0.8 |
| MMHC | 19.81±0.55 | 47.86±0.52 | 1.0±0.0 | 38.07±0.75 | 101.11±0.67 | 3.8±0.2 |
| MMHC-KKTS | 6.15±0.61 | 49.92±0.70 | 5.4±0.1 | 9.01±0.59 | 99.21±0.87 | 34.2±0.5 |
| PC | 22.46±0.67 | 61.68±0.63 | 3.1±0.1 | 40.36±0.73 | 124.45±0.82 | 4.6±0.1 |
| PC-KKTS | 3.81±0.39 | 49.57±0.69 | 7.6±0.1 | 5.39±0.46 | 98.62±0.84 | 33.1±0.5 |
| Search | 13.67±0.69 | 47.63±0.63 | 10.2±0.3 | 27.59±1.05 | 95.95±0.93 | 114.8±2.4 |
| CAM | 54.15±1.08 | 72.17±1.05 | 84.1±0.7 | 114.26±1.28 | 148.78±1.25 | 197.6±1.2 |
| CAM-KKTS | 9.71±0.85 | 50.35±0.72 | 88.4±0.7 | 20.48±1.21 | 101.37±0.94 | 229.1±1.2 |

Table 12: Results (mean ± standard error over 100 trials) on ER4 graphs with exponential noise, $n = 1000$.

| | $d = 10$ | | | $d = 30$ | | |
|---|---|---|---|---|---|---|
| | SHD | nnz | time (sec) | SHD | nnz | time (sec) |
| NOTEARS | 2.92±0.27 | 18.05±0.35 | 2.9±0.2 | 6.21±0.64 | 58.41±0.77 | 37.8±2.6 |
| NOTEARS-KKTS | 1.80±0.19 | 18.28±0.34 | 3.1±0.2 | 3.55±0.43 | 58.11±0.76 | 41.5±2.6 |
| NOTEARS-1e-5 | 3.07±0.28 | 17.82±0.35 | 2.0±0.1 | 6.97±0.61 | 57.67±0.77 | 15.4±0.8 |
| NOTEARS-1e-5-KKTS | 1.78±0.18 | 18.26±0.34 | 2.2±0.1 | 4.16±0.41 | 58.16±0.76 | 19.1±0.8 |
| Abs | 4.07±0.34 | 18.17±0.38 | 1.5±0.1 | 13.49±0.93 | 60.12±0.89 | 9.4±0.6 |
| Abs-KKTS | 2.24±0.25 | 18.29±0.34 | 1.8±0.1 | 4.99±0.53 | 57.96±0.77 | 13.2±0.7 |
| FGS | 13.20±0.81 | 29.51±0.98 | 0.5±0.0 | 49.28±3.29 | 113.40±4.46 | 1.2±0.1 |
| FGS-KKTS | 4.43±0.44 | 17.85±0.34 | 0.7±0.0 | 11.38±1.27 | 58.88±0.82 | 4.3±0.1 |
| MMHC | 14.81±0.53 | 12.04±0.19 | 0.5±0.0 | 41.11±1.13 | 36.41±0.42 | 2.1±0.0 |
| MMHC-KKTS | 6.01±0.52 | 17.35±0.34 | 0.6±0.0 | 21.05±1.39 | 59.09±0.92 | 4.2±0.0 |
| PC | 15.77±0.53 | 15.21±0.23 | 2.2±0.0 | 46.57±1.19 | 45.90±0.41 | 3.1±0.0 |
| PC-KKTS | 5.06±0.43 | 17.91±0.34 | 2.4±0.0 | 19.61±1.42 | 59.16±0.88 | 5.3±0.1 |
| Search | 8.68±0.60 | 15.94±0.35 | 0.5±0.0 | 31.58±1.39 | 52.62±0.70 | 6.9±0.2 |
| CAM | 20.12±0.52 | 23.58±0.41 | 10.0±0.1 | 67.48±1.65 | 82.05±1.14 | 45.0±0.6 |
| CAM-KKTS | 7.08±0.59 | 17.08±0.32 | 10.1±0.1 | 24.71±1.60 | 58.63±0.82 | 46.9±0.6 |

| | $d = 50$ | | | $d = 100$ | | |
|---|---|---|---|---|---|---|
| | SHD | nnz | time (sec) | SHD | nnz | time (sec) |
| NOTEARS | 13.10±1.30 | 97.14±0.94 | 149.3±7.7 | 21.62±1.31 | 196.95±1.37 | 660.6±20.2 |
| NOTEARS-KKTS | 7.80±0.81 | 97.17±0.88 | 165.0±7.8 | 11.89±0.89 | 197.80±1.32 | 787.8±20.5 |
| NOTEARS-1e-5 | 14.00±1.25 | 96.64±0.95 | 52.4±2.7 | 22.49±1.25 | 195.75±1.40 | 270.1±9.3 |
| NOTEARS-1e-5-KKTS | 8.35±0.85 | 97.24±0.88 | 68.1±2.8 | 12.81±0.88 | 197.48±1.33 | 396.6±9.6 |
| Abs | 28.00±1.61 | 101.50±1.19 | 33.4±2.1 | 54.99±2.39 | 208.17±1.99 | 193.2±9.6 |
| Abs-KKTS | 12.10±1.03 | 97.58±0.92 | 48.9±2.1 | 25.86±1.67 | 199.68±1.59 | 317.3±9.7 |
| FGS | 77.55±5.13 | 184.44±6.80 | 2.2±0.1 | 117.42±7.33 | 321.85±9.72 | 4.4±0.2 |
| FGS-KKTS | 21.17±1.90 | 99.95±1.12 | 15.1±0.3 | 36.27±3.03 | 206.12±1.99 | 119.4±2.5 |
| MMHC | 64.34±1.47 | 60.67±0.60 | 1.5±0.0 | 118.32±2.12 | 128.96±1.10 | 6.8±2.7 |
| MMHC-KKTS | 38.44±2.22 | 101.47±1.21 | 10.2±0.1 | 73.86±4.18 | 213.01±2.28 | 80.6±2.9 |
| PC | 72.39±1.54 | 75.24±0.59 | 4.5±0.1 | 130.04±2.49 | 154.14±0.92 | 8.7±0.1 |
| PC-KKTS | 36.55±2.16 | 102.24±1.26 | 13.4±0.2 | 64.06±3.88 | 210.39±2.26 | 73.3±1.0 |
| Search | 53.39±1.73 | 86.99±1.03 | 31.9±0.6 | 103.06±2.46 | 179.18±1.36 | 448.1±5.9 |
| CAM | 104.86±1.89 | 134.03±1.41 | 86.2±1.0 | 199.41±2.53 | 261.46±1.67 | 203.4±1.4 |
| CAM-KKTS | 41.92±2.29 | 102.43±1.15 | 94.2±1.0 | 85.55±3.39 | 213.69±1.91 | 273.9±2.0 |

Table 13: Results (mean $\pm$ standard error over 100 trials) on SF4 graphs with exponential noise, $n = 1000$.

| | $d = 10$ | | | $d = 30$ | | |
|---|---|---|---|---|---|---|
| | SHD | nnz | time (sec) | SHD | nnz | time (sec) |
| NOTEARS | 0.62±0.12 | 13.68±0.17 | 0.7±0.0 | 1.21±0.21 | 49.70±0.38 | 4.8±0.2 |
| NOTEARS-KKTS | 0.41±0.08 | 13.66±0.16 | 0.8±0.0 | 0.98±0.14 | 49.61±0.35 | 7.3±0.2 |
| NOTEARS-1e-5 | 0.63±0.12 | 13.67±0.17 | 0.5±0.0 | 1.24±0.21 | 49.67±0.38 | 3.3±0.2 |
| NOTEARS-1e-5-KKTS | 0.41±0.08 | 13.66±0.16 | 0.6±0.0 | 0.98±0.14 | 49.61±0.35 | 5.7±0.2 |
| Abs | 0.66±0.11 | 13.66±0.17 | 0.3±0.0 | 3.21±0.44 | 49.90±0.43 | 2.2±0.2 |
| Abs-KKTS | 0.47±0.08 | 13.63±0.16 | 0.4±0.0 | 1.52±0.28 | 49.52±0.36 | 4.8±0.2 |
| FGS | 4.69±0.57 | 17.21±0.66 | 0.4±0.0 | 21.93±1.75 | 64.47±2.22 | 0.8±0.0 |
| FGS-KKTS | 0.92±0.20 | 13.63±0.17 | 0.6±0.0 | 2.37±0.46 | 49.72±0.37 | 3.3±0.0 |
| MMHC | 6.66±0.38 | 11.28±0.12 | 0.5±0.0 | 28.25±0.67 | 33.41±0.43 | 13.0±5.0 |
| MMHC-KKTS | 1.51±0.23 | 13.47±0.17 | 0.6±0.0 | 4.10±0.56 | 49.46±0.40 | 14.7±5.0 |
| PC | 6.84±0.33 | 12.52±0.16 | 3.1±0.1 | 29.71±0.73 | 35.75±0.43 | 25.8±12.2 |
| PC-KKTS | 1.03±0.20 | 13.88±0.17 | 3.2±0.1 | 3.90±0.57 | 48.88±0.38 | 28.2±12.2 |
| Search | 2.26±0.28 | 13.13±0.20 | 0.3±0.0 | 11.28±1.06 | 48.86±0.41 | 3.7±0.1 |
| CAM | 20.58±0.62 | 23.29±0.42 | 10.5±0.2 | 66.45±1.33 | 65.58±1.03 | 47.3±0.7 |
| CAM-KKTS | 2.02±0.28 | 13.26±0.18 | 10.6±0.2 | 7.03±0.81 | 49.63±0.43 | 49.0±0.7 |

| | $d = 50$ | | | $d = 100$ | | |
|---|---|---|---|---|---|---|
| | SHD | nnz | time (sec) | SHD | nnz | time (sec) |
| NOTEARS | 1.60±0.23 | 87.37±0.42 | 16.6±0.8 | 2.91±0.57 | 183.84±0.58 | 122.3±6.1 |
| NOTEARS-KKTS | 1.37±0.23 | 87.44±0.41 | 26.8±0.9 | 2.51±0.52 | 183.83±0.58 | 210.7±6.5 |
| NOTEARS-1e-5 | 1.62±0.23 | 87.33±0.41 | 10.3±0.5 | 2.90±0.56 | 183.78±0.57 | 66.8±4.0 |
| NOTEARS-1e-5-KKTS | 1.38±0.23 | 87.43±0.41 | 20.7±0.6 | 2.51±0.52 | 183.83±0.58 | 152.7±4.5 |
| Abs | 6.44±1.06 | 88.63±0.71 | 7.6±0.6 | 16.37±1.68 | 187.11±0.98 | 59.2±5.4 |
| Abs-KKTS | 2.75±0.54 | 87.66±0.51 | 17.8±0.7 | 6.75±0.98 | 183.86±0.74 | 147.5±5.7 |
| FGS | 42.04±2.52 | 105.22±2.87 | 1.1±0.0 | 94.46±4.56 | 197.61±5.14 | 2.4±0.1 |
| FGS-KKTS | 7.41±1.28 | 87.88±0.53 | 11.7±0.1 | 19.20±2.68 | 186.21±1.43 | 93.5±1.0 |
| MMHC | 54.17±1.01 | 54.60±0.71 | 119.4±66.9 | 122.83±1.35 | 99.62±0.97 | 61.9±43.5 |
| MMHC-KKTS | 10.23±1.45 | 86.82±0.61 | 127.2±66.9 | 26.85±2.69 | 181.25±1.08 | 121.9±43.5 |
| PC | 55.54±0.81 | 59.18±0.67 | 16.0±2.9 | 121.91±1.22 | 120.34±0.85 | 24.5±3.8 |
| PC-KKTS | 8.11±0.98 | 86.61±0.54 | 25.1±2.8 | 19.18±1.51 | 179.10±0.75 | 101.3±3.9 |
| Search | 22.98±2.00 | 85.41±0.73 | 14.5±0.3 | 51.99±3.33 | 181.09±1.14 | 160.7±2.9 |
| CAM | 112.37±1.77 | 104.55±1.53 | 89.8±1.0 | 230.42±2.11 | 195.68±1.83 | 212.5±2.1 |
| CAM-KKTS | 16.02±1.66 | 88.90±0.69 | 97.2±1.0 | 33.75±2.47 | 185.03±1.30 | 275.2±2.5 |

Table 14: Results (mean ± standard error over 100 trials) on ER2 graphs with Gaussian noise, $n = 2d$.

| | $d = 10$ | | | $d = 30$ | | |
|---|---|---|---|---|---|---|
| | SHD | nnz | time (sec) | SHD | nnz | time (sec) |
| NOTEARS | 5.40±0.31 | 12.40±0.31 | 1.9±0.1 | 3.75±0.24 | 29.65±0.53 | 22.6±0.8 |
| NOTEARS-KKTS | 5.32±0.29 | 12.86±0.32 | 2.0±0.1 | 3.48±0.25 | 29.96±0.54 | 25.2±0.8 |
| NOTEARS-1e-5 | 5.40±0.32 | 12.33±0.31 | 0.9±0.0 | 3.94±0.29 | 29.45±0.53 | 4.8±0.3 |
| NOTEARS-1e-5-KKTS | 5.29±0.29 | 12.85±0.32 | 1.0±0.0 | 3.52±0.24 | 29.77±0.52 | 7.4±0.3 |
| Abs | 5.37±0.30 | 12.47±0.31 | 0.4±0.0 | 6.20±0.39 | 29.84±0.63 | 2.1±0.1 |
| Abs-KKTS | 5.42±0.28 | 12.85±0.31 | 0.6±0.0 | 4.21±0.31 | 29.76±0.56 | 4.7±0.1 |
| FGS | 15.27±0.44 | 25.78±0.60 | 0.4±0.0 | 16.04±0.89 | 30.28±0.82 | 0.5±0.0 |
| FGS-KKTS | 6.09±0.32 | 12.37±0.29 | 0.5±0.0 | 6.95±0.54 | 30.28±0.58 | 3.0±0.0 |
| MMHC | 8.78±0.33 | 5.01±0.12 | 0.4±0.0 | 20.03±0.54 | 21.90±0.28 | 0.5±0.0 |
| MMHC-KKTS | 6.71±0.37 | 11.98±0.35 | 0.5±0.0 | 9.72±0.70 | 30.35±0.62 | 2.1±0.0 |
| PC | 10.45±0.33 | 6.62±0.21 | 2.7±0.1 | 26.29±0.58 | 30.87±0.42 | 2.4±0.0 |
| PC-KKTS | 6.97±0.38 | 11.73±0.31 | 2.8±0.1 | 9.66±0.71 | 29.66±0.60 | 4.1±0.1 |
| Search | 6.56±0.37 | 11.77±0.28 | 0.3±0.0 | 11.59±0.70 | 29.04±0.55 | 5.9±0.1 |

| | $d = 50$ | | | $d = 100$ | | |
|---|---|---|---|---|---|---|
| | SHD | nnz | time (sec) | SHD | nnz | time (sec) |
| NOTEARS | 3.88±0.32 | 49.32±0.69 | 60.7±1.9 | 5.06±0.49 | 96.05±0.90 | 222.0±7.2 |
| NOTEARS-KKTS | 2.74±0.22 | 49.25±0.66 | 70.9±1.9 | 2.92±0.22 | 96.41±0.90 | 289.8±7.4 |
| NOTEARS-1e-5 | 4.42±0.33 | 48.94±0.67 | 12.4±0.6 | 5.28±0.50 | 95.84±0.92 | 66.3±2.8 |
| NOTEARS-1e-5-KKTS | 2.95±0.22 | 49.15±0.67 | 22.1±0.6 | 2.88±0.20 | 96.36±0.89 | 131.8±2.9 |
| Abs | 9.23±0.56 | 49.88±0.77 | 5.2±0.3 | 18.05±0.76 | 95.52±1.14 | 26.5±1.4 |
| Abs-KKTS | 3.93±0.30 | 49.00±0.65 | 15.3±0.3 | 6.18±0.40 | 95.84±0.93 | 95.4±1.6 |
| FGS | 20.92±0.70 | 52.16±0.89 | 0.6±0.0 | 31.17±0.82 | 103.36±1.16 | 0.9±0.0 |
| FGS-KKTS | 6.86±0.65 | 49.64±0.76 | 10.4±0.1 | 6.82±0.56 | 96.95±0.93 | 66.7±0.7 |
| MMHC | 29.96±0.70 | 34.56±0.37 | 0.7±0.0 | 50.80±0.87 | 79.48±0.63 | 4.5±0.2 |
| MMHC-KKTS | 14.35±0.96 | 50.08±0.78 | 6.4±0.1 | 19.22±1.16 | 97.61±1.05 | 45.9±0.5 |
| PC | 39.35±0.79 | 55.57±0.62 | 2.5±0.0 | 61.35±1.08 | 117.41±0.88 | 3.9±0.1 |
| PC-KKTS | 11.29±0.71 | 49.58±0.71 | 8.9±0.1 | 14.75±1.00 | 96.78±1.03 | 66.6±1.0 |
| Search | 18.11±0.82 | 48.58±0.67 | 26.8±0.2 | 31.22±1.22 | 94.18±0.99 | 282.7±3.0 |

Table 15: Results (mean $\pm$ standard error over 100 trials) on ER4 graphs with Gumbel noise, $n = 2d$.

| | $d = 10$ | | | $d = 30$ | | |
|---|---|---|---|---|---|---|
| | SHD | nnz | time (sec) | SHD | nnz | time (sec) |
| NOTEARS | 9.49±0.40 | 21.11±0.35 | 2.9±0.2 | 17.19±1.01 | 64.34±0.90 | 50.3±2.4 |
| NOTEARS-KKTS | 9.16±0.37 | 21.84±0.32 | 3.1±0.2 | 13.63±0.63 | 63.82±0.77 | 53.9±2.5 |
| NOTEARS-1e-5 | 9.43±0.40 | 20.95±0.35 | 2.1±0.1 | 18.56±1.05 | 64.10±0.90 | 18.5±0.8 |
| NOTEARS-1e-5-KKTS | 9.12±0.38 | 21.84±0.31 | 2.2±0.1 | 14.00±0.70 | 64.12±0.86 | 21.9±0.8 |
| Abs | 9.37±0.41 | 21.11±0.33 | 0.9±0.1 | 23.71±1.20 | 67.86±0.98 | 8.2±0.5 |
| Abs-KKTS | 8.68±0.32 | 21.82±0.32 | 1.1±0.1 | 16.51±0.90 | 64.77±0.84 | 11.4±0.5 |
| FGS | 20.34±0.47 | 32.84±0.56 | 0.4±0.0 | 50.57±1.54 | 64.90±1.33 | 0.6±0.0 |
| FGS-KKTS | 11.25±0.43 | 20.44±0.33 | 0.5±0.0 | 32.61±1.66 | 67.68±1.05 | 3.7±0.1 |
| MMHC | 17.81±0.37 | 5.74±0.12 | 0.4±0.0 | 50.29±0.86 | 23.78±0.28 | 0.5±0.0 |
| MMHC-KKTS | 12.20±0.51 | 19.90±0.35 | 0.5±0.0 | 43.91±1.73 | 69.74±1.12 | 2.4±0.0 |
| PC | 20.10±0.34 | 7.22±0.26 | 2.2±0.0 | 58.42±0.86 | 35.84±0.45 | 2.3±0.0 |
| PC-KKTS | 13.24±0.47 | 19.56±0.40 | 2.4±0.0 | 42.51±1.65 | 68.63±1.15 | 4.2±0.0 |
| Search | 13.37±0.54 | 17.81±0.40 | 0.4±0.0 | 42.55±1.16 | 55.10±0.73 | 8.4±0.1 |

| | $d = 50$ | | | $d = 100$ | | |
|---|---|---|---|---|---|---|
| | SHD | nnz | time (sec) | SHD | nnz | time (sec) |
| NOTEARS | 20.49±1.39 | 102.65±1.26 | 156.3±3.8 | 30.60±1.89 | 202.11±1.60 | 660.5±10.3 |
| NOTEARS-KKTS | 11.38±0.76 | 100.56±1.09 | 171.1±3.8 | 16.17±1.29 | 200.32±1.52 | 771.9±10.4 |
| NOTEARS-1e-5 | 21.26±1.29 | 101.84±1.19 | 52.3±1.7 | 30.92±1.89 | 201.33±1.63 | 265.0±9.0 |
| NOTEARS-1e-5-KKTS | 13.37±0.95 | 101.33±1.15 | 66.2±1.7 | 16.47±1.27 | 200.17±1.50 | 375.2±9.2 |
| Abs | 33.02±1.93 | 109.21±1.56 | 28.9±1.8 | 55.70±2.25 | 214.18±1.93 | 142.7±7.7 |
| Abs-KKTS | 17.00±1.14 | 102.61±1.20 | 43.2±1.9 | 25.22±1.55 | 203.79±1.71 | 253.4±7.7 |
| FGS | 76.30±2.29 | 121.86±2.14 | 1.0±0.0 | 108.31±3.42 | 246.38±3.90 | 2.0±0.0 |
| FGS-KKTS | 46.87±2.65 | 113.11±1.74 | 13.4±0.3 | 57.28±3.40 | 214.57±2.30 | 104.5±1.9 |
| MMHC | 78.41±1.34 | 44.90±0.49 | 1.6±0.0 | 141.90±1.91 | 106.56±0.73 | 8.4±0.1 |
| MMHC-KKTS | 69.17±3.13 | 118.96±1.79 | 9.0±0.1 | 118.61±4.77 | 238.61±3.20 | 84.0±1.4 |
| PC | 93.50±1.33 | 66.67±0.64 | 2.5±0.0 | 163.85±2.18 | 140.41±1.03 | 4.3±0.1 |
| PC-KKTS | 74.18±3.03 | 120.62±1.94 | 9.8±0.1 | 124.97±5.04 | 239.79±3.10 | 63.0±0.7 |
| Search | 65.70±2.02 | 94.82±1.07 | 47.1±0.5 | 111.35±2.53 | 190.97±1.42 | 700.1±4.3 |

Table 16: Results (mean $\pm$ standard error over 100 trials) on SF4 graphs with exponential noise, $n = 2d$.

| | $d = 10$ | | | $d = 30$ | | |
|---|---|---|---|---|---|---|
| | SHD | nnz | time (sec) | SHD | nnz | time (sec) |
| NOTEARS | 7.02±0.35 | 15.39±0.27 | 1.3±0.1 | 7.63±0.37 | 50.74±0.49 | 17.5±0.3 |
| NOTEARS-KKTS | 6.85±0.34 | 15.54±0.26 | 1.5±0.1 | 7.27±0.34 | 50.44±0.45 | 20.1±0.4 |
| NOTEARS-1e-5 | 6.95±0.35 | 15.26±0.27 | 0.8±0.0 | 7.50±0.36 | 50.48±0.49 | 4.8±0.1 |
| NOTEARS-1e-5-KKTS | 6.82±0.34 | 15.53±0.26 | 0.9±0.0 | 7.25±0.34 | 50.30±0.44 | 7.4±0.2 |
| Abs | 6.30±0.32 | 15.18±0.26 | 0.3±0.0 | 9.37±0.52 | 50.30±0.57 | 2.3±0.1 |
| Abs-KKTS | 6.70±0.34 | 15.49±0.26 | 0.5±0.0 | 7.83±0.41 | 50.14±0.45 | 4.9±0.1 |
| FGS | 17.57±0.55 | 29.69±0.70 | 0.3±0.0 | 37.53±0.55 | 28.26±0.56 | 0.5±0.0 |
| FGS-KKTS | 7.11±0.34 | 15.13±0.27 | 0.4±0.0 | 14.93±0.89 | 50.26±0.58 | 3.2±0.0 |
| MMHC | 12.49±0.22 | 4.16±0.13 | 0.4±0.0 | 43.18±0.42 | 15.28±0.25 | 0.5±0.0 |
| MMHC-KKTS | 7.67±0.34 | 14.64±0.29 | 0.5±0.0 | 18.58±1.19 | 50.29±0.71 | 2.1±0.0 |
| PC | 13.96±0.16 | 5.82±0.21 | 2.2±0.0 | 47.17±0.44 | 24.56±0.41 | 2.2±0.0 |
| PC-KKTS | 7.19±0.30 | 14.74±0.25 | 2.3±0.0 | 17.10±0.90 | 51.20±0.54 | 3.8±0.0 |
| Search | 7.95±0.35 | 13.95±0.27 | 0.3±0.0 | 19.55±1.09 | 47.54±0.53 | 4.8±0.1 |

| | $d = 50$ | | | $d = 100$ | | |
|---|---|---|---|---|---|---|
| | SHD | nnz | time (sec) | SHD | nnz | time (sec) |
| NOTEARS | 5.94±0.29 | 86.62±0.48 | 49.7±0.9 | 6.91±0.65 | 182.18±0.65 | 203.0±3.6 |
| NOTEARS-KKTS | 5.98±0.33 | 86.69±0.49 | 60.1±0.9 | 6.12±0.61 | 182.02±0.64 | 281.6±4.0 |
| NOTEARS-1e-5 | 6.04±0.29 | 86.52±0.48 | 12.5±0.5 | 6.97±0.65 | 182.15±0.65 | 73.7±2.7 |
| NOTEARS-1e-5-KKTS | 5.94±0.33 | 86.56±0.49 | 22.7±0.6 | 6.15±0.61 | 181.99±0.64 | 150.5±3.1 |
| Abs | 13.46±1.33 | 87.19±0.81 | 7.0±0.5 | 19.81±1.44 | 182.82±1.03 | 48.4±3.4 |
| Abs-KKTS | 9.41±0.86 | 86.80±0.60 | 17.2±0.5 | 11.04±1.12 | 181.49±0.69 | 123.3±3.7 |
| FGS | 63.93±0.80 | 50.98±0.90 | 0.6±0.0 | 121.68±1.82 | 119.36±1.80 | 1.0±0.0 |
| FGS-KKTS | 22.54±1.70 | 86.05±0.82 | 10.9±0.1 | 50.50±3.58 | 180.72±1.70 | 75.4±1.0 |
| MMHC | 73.67±0.56 | 29.46±0.34 | 1.5±0.0 | 145.51±1.12 | 74.83±0.62 | 5.4±0.0 |
| MMHC-KKTS | 31.46±2.11 | 86.52±0.89 | 7.3±0.1 | 75.14±4.37 | 181.41±1.49 | 55.4±0.6 |
| PC | 77.16±0.57 | 45.53±0.53 | 2.3±0.0 | 146.92±1.05 | 100.22±0.71 | 3.4±0.1 |
| PC-KKTS | 35.87±2.20 | 88.69±0.85 | 8.2±0.1 | 66.27±4.40 | 177.25±1.60 | 43.5±0.5 |
| Search | 28.94±1.77 | 81.59±0.67 | 23.5±0.3 | 57.61±3.44 | 173.72±1.00 | 253.1±2.6 |