[Reviews · NeurIPS 2020]

Review 1

Summary and Contributions: The paper shows that the NOTEARS approach to learning linear SEMs rarely meets KKT conditions. Also alternative approaches to ruling out cycles (absolute value) are considered. Informed by theoretical connections between KKT conditions and edge absences, a local search method is presented (which can use a NOTEARS solution as a starting point). This method improves on plain NOTEARS.

Strengths: Useful negative results are presented which inform a new and empirically successful algorithm. The NOTEARS acyclicity constraints are generalized (and empirically assessed). The authors are forthright about what still remains unclear.

Weaknesses: It is difficult to work out what leads to better or worse SHD results. We have choices for algorithm but also F (always squared error here), thresholding, acyclicity constraint and even centering (which makes a surprisingly big difference). Also the results for plain NOTEARS are different from an earlier paper, which is odd. Given that this is learning by constrained optimization it would be good to more clearly separate out (i) the effect of choice of objective (and constraints) and (ii) the effect of the failure to solve the constrained optimization problem to optimality (i.e. how good is our local optima?)

Correctness: Yes. At least I did not find any errors. Comparison is only to NOTEARS variants and GES (FGS implementation) and CAM, but we are assured that nothing else is worth considering. I will just have to trust the authors here.

Clarity: The paper stresses the "fully continuous" nature of the NOTEARS approach but with thresholding I think this is an overstatement. The -KKTS methods presented here are not "fully continuous". It doesn't matter whether or not we have a fully continuous method, we just care whether it works.

Relation to Prior Work: The whole paper is about a development of the basic NOTEARS approach.

Reproducibility: Yes

Additional Feedback: It is written: "This section analyzes the continuous optimization problem of minimizing a score function F (W ) subject to the acyclicity constraint h(A) = 0 for any h defined by (1)". We see in the supplementary material that in the experiments presented here "least-squares loss" is always used as the "F". Is there not a danger of overfitting here? Perhaps because both NOTEARS and NOTEARS-KKTS only find local optima, this optimization failure is actually helpful and is providing a kind of overfitting-avoidance. Or perhaps somehow working out a good threshold value (here 0.3) to convert weights into edges is what matters. AFTER DISCUSSION/FEEDBACK Thanks to the authors for their feedback. This paper indeed raises a lot of as-yet-unanswered questions. This is perhaps not such a bad thing.


Review 2

Summary and Contributions: The paper builds on an existing method for learning Bayesian networks from data (NOTEARS) using continuous optimisation. The paper argues that there are theoretical problems with convergence to optimality in the existing method and derives a related method that does not have this problem. Experimental evaluation is presented. This shows that the novel method is actually rather poor in practice - however, as part of a hybrid algorithm, it outperforms the existing state-of-the-art.

Strengths: The paper provides a clear explanation of the theoretical problems with the existing NOTEARS algorithm, and explains how the method should be altered in order to remove this problem. The presented technique (when used as a post-processing step) gives a significant improvement in accuracy over the current techniques of this type. Results appear to be about a factor of 2 better in most cases. This is achieved without a huge increase in runtime. Some of the theoretical insight into the problem domain is a potentially useful side effect for others looking to develop alternative techniques in this area.

Weaknesses: The paper essentially fixes a problem with an existing approach. While this is definitely worthwhile and results in an algorithm that is both theoretically and practically superior, this is possibly less interesting than a completely novel technique.

Correctness: The theoretical results appear to be correct to the best of my ability to understand them. The experimental method followed is also technically sound - the fact that it largely replicates experimental settings already presented in the literature is a definite plus.

Clarity: The paper is written very clearly throughout. It is surprisingly readable for a paper with lots of theoretical results and the fact that the theorems involved are linked back to the graphs where possible greatly helps the understanding of the material.

Relation to Prior Work: The entire paper is based around issues with existing approaches and so the previous work in the area is properly explained and the novel contribution of this paper is clearly given. Other less directly related work is briefly mentioned but not really explored.

Reproducibility: Yes

Additional Feedback: The final proposed method for post-processing the results from the NOTEARS algorithm is reminiscent of hybrid Bayesian network learning techniques, where one algorithm is used to learn a network which is then used as the starting point for a local search by another algorithm (e.g. Max-Min Hill Climbing). It might be worth drawing this comparison. =================================== UPDATED - I read the other reviews and the authors' response. This has not substantially changed my opinion of the paper.


Review 3

Summary and Contributions: The paper advances the "NOTEARS method" for finding high-scoring Bayesian networks. Specifically, a reformulation is given so as to satisfy the KKT conditions for local optima. The development is shown to yield local search algorithms that can improve upon the performance of NOTEARS in terms of structural error (SHD) and computation time.

Strengths: + Structure learning in Bayesian networks is a challenging and very well motivated machine learning problem. + The paper improves the understanding of the strengths and weaknesses of the NOTEARS approach.

Weaknesses: - The paper is fairly incremental, developing a single heuristic local search method (namely NOTEARS that enjoys no non-trivial performance guarantees). - I find the claim that NOTEARS and FGS outperform earlier methods (Line 276) questionable. E.g., for moderate numbers of nodes d, as studied in the present paper, Ref. 2 (Table 1, p. 2306) shows that MMHC and PC perform much better than GES (here FGS). This is critical for the overall positioning of the empirical results in the context of the state of the art. (The same weakness concern also the original NOTEARS work and many follow-up works.)

Correctness: As said above, viewing NOTEARS as a representative of the state of the art is questionable.

Clarity: The paper is generally well written, given the limited number of pages.

Relation to Prior Work: The relation to NOTEARS (which is the main point of the work) is well discussed.

Reproducibility: Yes

Additional Feedback: -- Added after the discussion phase -- Thanks for the rebuttal. Let me clarify that, to me, achieving a local optimum does not count, as such, as a "non-trivial performance guarantee". That said, I realize that the paper does contribute an efficient means for significantly improving the empirical accuracy wrt SHD, not just within the somewhat narrow context of NOTEARS (local search heuristic for linear models with ad-hoc post-pruning instead of optimizing one principled scocing function), but also for other methods, such as GES, FGS, MMHC. It was most likely my mistake to think that FGS is just much faster (and thus less accurate) than the old GES; now I believe FGS is indeed also much more accurate than GES wrt SHD. Thanks for the additional results.


Review 4

Summary and Contributions: The paper introduces novel insights on the topic of learning BNs through a fully continuous optimization formulation.

Strengths: The introduced novel formulation is indeed interesting, and the experimental results are promising.

Weaknesses: The problem in the paper is that it fails in showing the actual scope of the new results, especially in the global context of BNs learning. In fact their methods apparently can only applied to the continuous case: no mention is ever made if the same method can work with categorical variables. This is reflected to the selected set of "state-of-the-art" methods against which they compare their methods, that is a narrow subset of the whole literature on BNs learning. Saying something like "As mentioned, this paper is most closely related to the fully continuous framework of ... " is definitely not enough: a more precise and thorough description of the limitations of this work, and its position in the whole BNs learning literature, is needed. The title and the abstract should modified as well with same reasoning.

Correctness: The methodology is sound.

Clarity: The paper is clear and succint. The language is correct.

Relation to Prior Work: The relation to previous work is clearly exposed.

Reproducibility: Yes

Additional Feedback:

[Author Response · NeurIPS 2020]

We thank the reviewers for their efforts. We are glad that R1, R2, R3 appreciated the improved understanding of
NOTEARS ("useful negative results"). Thanks to R2 for describing our theoretical insights in general as "a potentially
useful side effect for others" and "surprisingly readable for a paper with lots of theoretical results." R1, R2, R4 recognize
our empirical results to be "successful" and "a significant improvement." Below we respond to reviewer comments.

**R3: Questionable that NOTEARS, FGS outperform earlier methods, [2, Table 1] shows MMHC, PC perform**
**much better than GES (here FGS) for moderate** $d$**.** Thank you for pointing this out. We agree that the statement
on line 276 is too broad and will remove it. To address R3's concern, we first compared with MMHC and PC in the
experimental setting of Sec. 5. The significance level $\alpha$ was chosen from the range considered in [2] to minimize SHD.
The two left panels below show that while MMHC and PC do not perform better than FGS, they are also significantly
improved by KKTS (we will report full results in the paper). We then performed a second experiment with $n = 2d$ to
be closer to the setting of [2, Table 1], also adding the GES implementation used in [2]. The right panel shows that FGS
is actually an improvement over GES, remaining better than MMHC and PC (except $d = 10$) while GES is worse.

**R3: "Paper is fairly incremental, developing a single heuristic local search method (namely NOTEARS that**
**enjoys no non-trivial performance guarantees)."** We naturally disagree about incrementality and are not sure what is
meant by non-trivial. Prop. 3 provides a negative guarantee for NOTEARS (which is *not* our method), whereas Thms 9
and 7 provide positive guarantees for our KKTS method to yield KKT points and local minima. We would thus not call
KKTS heuristic or lacking guarantees. To get from Prop. 3 to KKTS requires several more contributions: reformulating
the problem and proving that KKT conditions are necessary (Thm 6), and relating the KKT conditions to edge absence
constraints (Lem. 6, Thm 8). Sec. 2 makes additional contributions in generalizing acyclicity constraints from [32,30].

**R4: Apparently only applicable to continuous case, no mention of categorical. More needed on limitations and**
**position in literature.** We thank R4 for prompting us to elaborate upon the problem setting and what remains for future
work, which we will do in the paper. The theory and methods apply straightforwardly to binary variables (although
we have not experimented with them) but not non-binary categorical variables. As Sec. 3, para. 1 states, the key
assumption is that each edge is associated with a single parameter $W_{ij}$. In a (generalized) linear structural equation, a
single parameter can account for the effect of a binary or continuous input, while a binary output can be handled by a
suitable loss function (e.g. logistic). However, a single parameter is likely insufficient for a non-binary categorical input
(typically encoded into multiple binary variables) or output (e.g. [14] proposes multi-logit regression with parameters
for each level). Therefore the extension to multiple parameters per edge (Sec. 6) is desirable to address categorical
variables as well as nonlinear models. **Abstract:** We will add a sentence on the one-parameter-per-edge assumption.
**Title:** We find it difficult to capture this assumption in a few readily understood words, but perhaps R4 has a suggestion.

**R1: "What leads to better or worse SHD...F (always squared error...danger of overfitting?), thresholding,**
**acyclicity constraint and even centering."** Thanks for the thoughtful questions. Below we summarize what we
know/have reported. We think a proper exploration would best be left to a journal extension of this paper. **Score**
**function** $F$**:** By keeping this as least squares, we have somewhat avoided overfitting to the noise type (Gaussian,
Gumbel, etc., usually unknown) as opposed to using the log-likelihood. [32, Sec. 5.3] shows that NOTEARS can
achieve scores close to those of the exact optimizer GOBNILP (especially before thresholding), i.e. it is fitting well
but not over-fitting, but we have not done a similar comparison for NOTEARS-KKTS. **Acyclicity constraint:** The
NOTEARS vs. Abs comparison shows that setting $A = W \circ W$ is empirically superior to $A = |W|$. As for the function
$h$, Appendix C.1 states that the polynomial $h(A)$ from [30] performs slightly better and is slightly faster than the
exponential $h(A)$ from [32] (the authors of [32] seem to agree as their code now uses polynomial $h$). Section 6 mentions
that other $h$ in the class of eq. (1) could be explored in future work. **Thresholding:** We followed [32] and fixed the
threshold $\omega = 0.3$ for NOTEARS and for our NOTEARS-inspired algorithms (Abs, KKTS), in part to demonstrate
success without too much parameter tuning. But we agree that the role of thresholding could be further explored.
**Centering:** We were also surprised by the effect this had, as reported in Appendix C.3.

**R1: "Fully continuous" is overstatement.** We will remove "fully". Note that NOTEARS [32] also uses thresholding.

[Meta-Review · NeurIPS 2020]

After discussions, there has been consensus that the paper's ideas deserve publication, even though they are somewhat incremental and without guarantees, as they build on NOTEARS. It has been appreciated the discussion on the issues with NOTEARS and an attempt to improve on them.